# Associations between mental health, blood pressure and the development of hypertension

H. Lina Schaare ®[1,2,3,4] ✉, Maria Blöchl[1,5], Deniz Kumral ®[1,6,7], Marie Uhlig[1], Lorenz Lemcke[8], Sofie L. Valk[2,3,4] & Arno Villringer ®[1,9,10,11]

Multiple studies have reported a link between mental health and high blood pressure with mixed or even contradictory findings. Here, we resolve those contradictions and further dissect the cross-sectional and longitudinal relationship between mental health, systolic blood pressure, and hypertension using extensive psychological, medical and neuroimaging data from the UK Biobank. We show that higher systolic blood pressure is associated with fewer depressive symptoms, greater well-being, and lower emotion-related brain activity. Interestingly, impending hypertension is associated with poorer mental health years before HTN is diagnosed. In addition, a stronger baseline association between systolic blood pressure and better mental health was observed in individuals who develop hypertension until follow-up. Overall, our findings offer insights on the complex relationship between mental health, blood pressure, and hypertension, suggesting that—via baroreceptor mechanisms and reinforcement learning—the association of higher blood pressure with better mental health may ultimately contribute to the development of hypertension.

Both hypertension (HTN) and affective disorders, such as depression, frequently co-occur and have been identified as single[1–4] as well as combined[5] risk factors for cardiovascular disease (CVD). An increased risk of HTN has been described in patients with affective disorders[6–10]. The burden of vascular risk factors, including HTN, has further been suggested to drive depressive symptoms in ageing through microvascular brain damage[11].

In contrast to these findings, some studies showed that higher blood pressure relates to better mood, higher well-being, and lower distress in healthy[12–16] and clinical populations[17–19]. Baroreceptor mechanisms have been suggested to explain these effects, as their intrinsic and experimentally-induced signalling has been shown to phasically adjust pain sensitivity thresholds, alter sensory and emotional processing, decrease cortical excitability, and inhibit central nervous system activity[20–26]. These observations have been proposed as a critical neuro-behavioural component in the development of essential HTN. Momentary relief from an adverse state might positively reinforce blood pressure-elevating behaviours and thus, via baroreceptor-mediated neural circuits, insidiously increase blood pressure over time, resulting in 'learned hypertension'[20,25,27–29]. However, it remains unclear if blood pressure elevations and HTN development relate to mental health and if such an association reflects in brain function.

[1]Department of Neurology, Max Planck Institute for Human Cognitive and Brain Sciences, Leipzig, Germany. [2]Otto-Hahn-Group Cognitive Neurogenetics, Max Planck Institute for Human Cognitive and Brain Sciences, Leipzig, Germany. [3]Institute of Neuroscience and Medicine (INM-7: Brain and Behaviour), Research Centre Jülich, Jülich, Germany. [4]Institute of Systems Neuroscience, Medical Faculty, Heinrich Heine University Düsseldorf, Düsseldorf, Germany. [5]Institute for Psychology, Leipzig University, Leipzig, Germany. [6]Institute of Psychology, Neuropsychology, University of Freiburg, Freiburg, Germany. [7]Institute of Psychology, Clinical Psychology and Psychotherapy Unit, University of Freiburg, Freiburg, Germany. [8]Nuclear Magnetic Resonance Unit, Max Planck Institute for Human Cognitive and Brain Sciences, Leipzig, Germany. [9]MindBrainBody Institute, Berlin School of Mind and Brain, Berlin, Germany. [10]Clinic of Cognitive Neurology, Leipzig University, Leipzig, Germany. [11]Charité University Medicine Berlin, Berlin, Germany. ✉e-mail: schaare@cbs.mpg.de

The first goal of the present study was to systematically describe the relationship of blood pressure with depressive symptoms and well-being, while accounting for potential confounding effects of medication intake and chronic illness, such as CVD and clinical depression. Due to small effect sizes reported in previous research related to our study[16–18], we capitalized on the unique study design offered by the UK Biobank. The UK Biobank combines a deeply phenotyped longitudinal cohort with high statistical power of more than 500,000 participants[30] which enables the detection of robust small effects. In addition, it includes two follow-up timepoints for longitudinal analyses in two sub-samples of the baseline cohort: an online mental health follow-up at around 5 years and a follow-up at around 10 years. We hypothesized for cross-sectional and longitudinal analyses that increased blood pressure relates to fewer depressive symptoms and greater well-being (preregistration: https://osf.io/638jg/).

The second goal of this study was to explore the relevance of blood pressure-mental health associations in relation to HTN development. If a systematic relationship between mental health and blood pressure exists, this could positively reinforce long-term blood pressure elevations via baroreceptor mechanisms. Hence, we explored if the relationship between mental health and blood pressure differed between participants who developed hypertension compared to those who stayed normotensive.

The third goal of this study was to explore the prospective effects of blood pressure-mental health associations in emotion-related brain function using UK Biobank's imaging assessment (N > 20,000). Due to the previously established baroreceptor effects on central nervous processing, blood pressure elevations and the development of HTN might relate to more general affective processes in brain function. We therefore investigated the effect of blood pressure variations on emotional task-based functional MRI activation[31,32].

## Results

### Participants

Overall, there were data of 502,494 participants (273,378 [54.4%] women) at initial assessment visit of whom 47,933 (24,793 [51.7%] women) participants attended the follow-up visit. (Fig. 1, Table 1). Average time between initial assessment and follow-up assessment was approximately 9 years (mean = 3261 days [8.9 years], range = 1400–5043 days [3.8–13.8 years]). At baseline, the median age was 58 years (range = 37–73 years) and 135,745 (27%) participants reported that they had previously been diagnosed with HTN (Table 1). The final sample size for each analysis after exclusion of missing data is reported in the respective results sections below.

### Descriptive data

As shown in Table 1, participants with and without HTN diagnosis differed with respect to all descriptive variables at initial assessment. Missing data were evident in all variables of interest with varying impact (SBP 45,540 [9.1%]), HTN 930 [0.2%], depressive symptoms 53,563 [10.7%], and well-being 385,271 [76.7%]). At follow-up assessment, missing data were evident for SBP (11,269 [23.5%]), HTN (36,264 [75.7%]), depressive symptoms (3285 [6.9%]), and well-being (371 [0.8%]). Missing data were listwise excluded and all following analyses

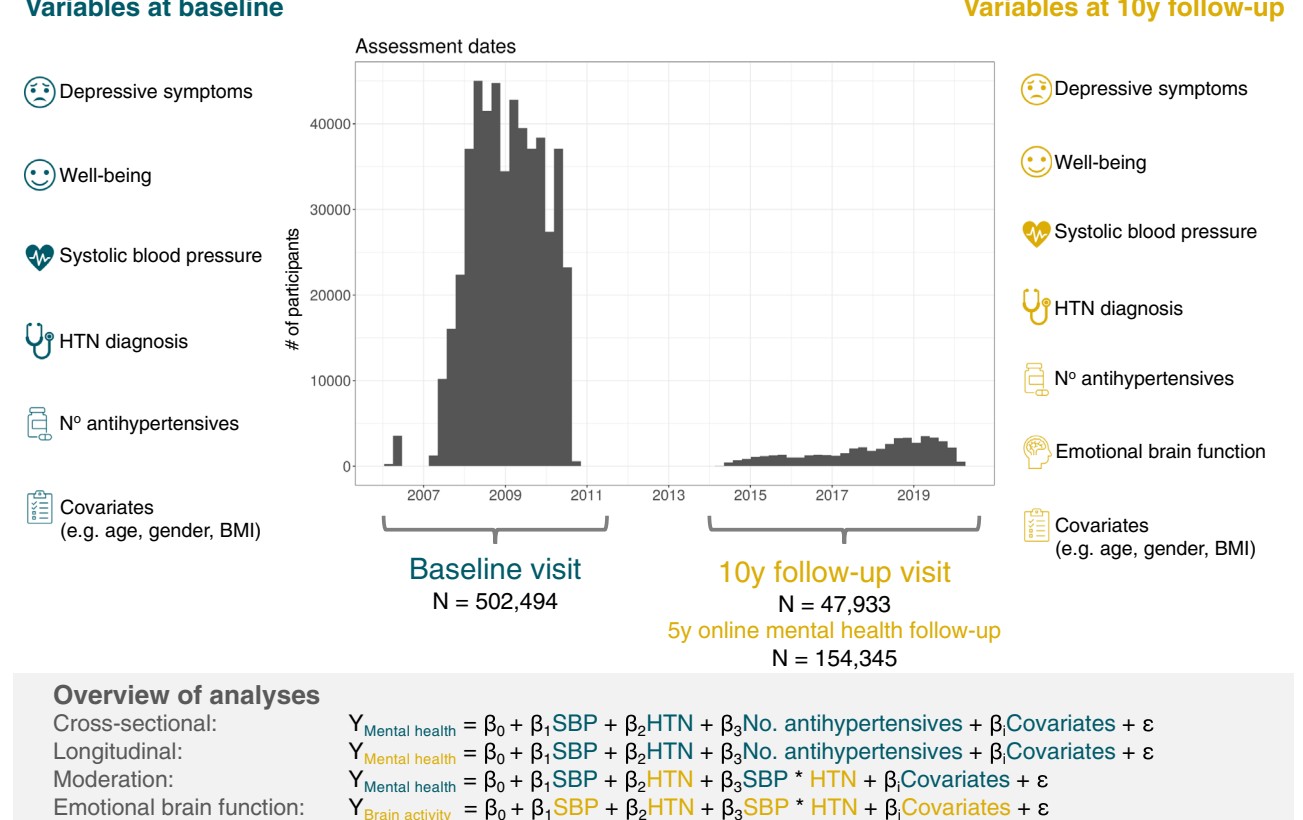

**Fig. 1 | Schematic overview of study design, outcome and predictor variables and analyses.** Our study included UK Biobank data (i) from the complete baseline assessment, which was conducted between 2006–2010, and (ii) from the ongoing 10-year follow-up visit which has started in 2014 and included brain magnetic resonance imaging (MRI). Our follow-up visit sample consisted of those participants whose data had been released by February 2020. In addition, a sub-sample of the baseline cohort participated in an online mental health follow-up assessment in 2016. We defined mental health variables (depressive symptoms, well-being) at baseline and follow-up as outcomes. Systolic blood pressure (SBP), hypertension diagnosis (HTN), and number of prescribed antihypertensive medications at baseline and follow-up were the main predictors in all models. Finally, we assessed the relationship of the main predictors with emotion-related brain function (i.e., BOLD fMRI response in Hariri task) at follow-up visit.

**Table 1 | Sample characteristics at baseline assessment for total sample, as well as sub-groups with and without diagnosed hypertension (HTN)**

| | | Overall (N = 502494) | No diagnosed HTN or unknown (N = 365819) | Diagnosed HTN (N = 135745) | P-value | Effect size |
|---|---|---|---|---|---|---|
| **Gender** | Female | 273378 (54.4%) | 206978 (56.6%) | 65932 (48.6%) | 2.22e–16 | 0.05 |
| | Male | 229115 (45.6%) | 158841 (43.4%) | 69813 (51.4%) | | |
| | Missing | 1 (0.0%) | 0 (0%) | 0 (0%) | | |
| **Age (years)** | Mean (SD) | 56.5 (8.10) | 55.4 (8.18) | 59.5 (7.09) | 2.22e–16 | 0.508 |
| | Median [Min, Max] | 58.0 [37.0, 73.0] | 56.0 [38.0, 73.0] | 61.0 [39.0, 72.0] | | |
| | Missing | 1 (0.0%) | 0 (0%) | 0 (0%) | | |
| **Systolic blood pressure (mmHg)** | Mean (SD) | 138 (18.6) | 134 (17.6) | 147 (18.1) | 2.22e–16 | 0.725 |
| | Median [Min, Max] | 136 [65.0, 254] | 133 [65.0, 253] | 146 [79.0, 254] | | |
| | Missing | 45540 (9.1%) | 32394 (8.9%) | 12737 (9.4%) | | |
| **Diastolic blood pressure (mmHg)** | Mean (SD) | 82.2 (10.1) | 80.7 (9.69) | 86.2 (10.2) | 2.22e–16 | 0.56 |
| | Median [Min, Max] | 82.0 [36.5, 148] | 80.5 [36.5, 140] | 86.0 [43.5, 148] | | |
| | Missing | 45528 (9.1%) | 32387 (8.9%) | 12732 (9.4%) | | |
| **Heart rate (beats/min)** | Mean (SD) | 69.3 (11.2) | 68.7 (10.7) | 70.9 (12.4) | 2.22e–16 | 0.192 |
| | Median [Min, Max] | 68.5 [30.5, 173] | 68.0 [30.5, 173] | 70.0 [30.5, 170] | | |
| | Missing | 45528 (9.1%) | 32387 (8.9%) | 12732 (9.4%) | | |
| **BMI (kg/m$^2$)** | Mean (SD) | 27.4 (4.80) | 26.7 (4.41) | 29.4 (5.25) | 2.22e–16 | 0.584 |
| | Median [Min, Max] | 26.7 [12.1, 74.7] | 26.1 [12.1, 69.0] | 28.6 [13.8, 74.7] | | |
| | Missing | 3105 (0.6%) | 1780 (0.5%) | 928 (0.7%) | | |
| **Diabetes** | Prefer not to answer / Do not know | 404 (0.1%) / 1280 (0.3%) | 347 (0.1%) / 726 (0.2%) | 57 (0.0%) | | |
| | No | 473479 (94.2%) | 354961 (97.0%) | 118518 (87.3%) | 2.22e–16 | 0.135 |
| | Yes | 26399 (5.3%) | 9785 (2.7%) | 16614 (12.2%) | | |
| | Missing | 932 (0.2%) | 0 (0%) | 2 (0.0%) | | |
| **Angina** | No diagnosed angina or unknown | 358910 (71.4%) | 233569 (63.8%) | 124955 (92.1%) | 2.22e–16 | 0.06 |
| | Diagnosed angina | 16117 (3.2%) | 6735 (1.8%) | 9370 (6.9%) | | |
| | Missing | 127467 (25.4%) | 125515 (34.3%) | 1420 (1.0%) | | |
| **Heart attack** | No diagnosed heart attack or unknown | 363524 (72.3%) | 234891 (64.2%) | 128249 (94.5%) | 2.22e–16 | 0.039 |
| | Diagnosed heart attack | 11503 (2.3%) | 5413 (1.5%) | 6076 (4.5%) | | |
| | Missing | 127467 (25.4%) | 125515 (34.3%) | 1420 (1.0%) | | |
| **Lifetime depression** | No diagnosed depression or unknown | 346919 (69.0%) | 220783 (60.4%) | 125778 (92.7%) | 2.22e–16 | 0.02 |
| | Diagnosed depression | 28108 (5.6%) | 19521 (5.3%) | 8547 (6.3%) | | |
| | Missing | 127467 (25.4%) | 125515 (34.3%) | 1420 (1.0%) | | |
| **No. antihypertensive medication** | Mean (SD) | 0.403 (0.864) | 0.0842 (0.405) | 1.26 (1.14) | 2.22e–16 | 1.717 |
| | Median [Min, Max] | 0.00 [0.00, 9.00] | 0.00 [0.00, 8.00] | 1.00 [0.00, 9.00] | | |
| **No. antidepressant medication** | Mean (SD) | 0.0784 (0.281) | 0.0700 (0.266) | 0.101 (0.318) | 2.22e–16 | 0.111 |
| | Median [Min, Max] | 0.00 [0.00, 5.00] | 0.00 [0.00, 3.00] | 0.00 [0.00, 5.00] | | |
| **Current depressive symptoms** | Mean (SD) | 1.40 (0.528) | 1.39 (0.513) | 1.44 (0.564) | 2.22e–16 | 0.106 |
| | Median [Min, Max] | 1.25 [1.00, 4.00] | 1.25 [1.00, 4.00] | 1.25 [1.00, 4.00] | | |
| | Missing | 53563 (10.7%) | 36900 (10.1%) | 15745 (11.6%) | | |
| **Well-being** | Mean (SD) | 4.46 (0.579) | 4.48 (0.571) | 4.41 (0.598) | 2.22e–16 | 0.133 |
| | Median [Min, Max] | 4.50 [1.00, 6.00] | 4.50 [1.00, 6.00] | 4.40 [1.00, 6.00] | | |
| | Missing | 330042 (65.7%) | 240332 (65.7%) | 88780 (65.4%) | | |
| **Diagnosed hypertension** | No diagnosed HTN or unknown | 365819 (72.8%) | 365819 (100%) | 0 (0%) | – | – |
| | Diagnosed HTN | 135745 (27.0%) | 0 (0%) | 135745 (100%) | | |
| | Missing | 930 (0.2%) | 0 (0%) | 0 (0%) | | |

*P-values and effect sizes refer to the comparison of HTN sub-groups for the respective variable using two-sided two sample t-tests for interval scale variables and Chi-square tests for nominal scale variables. Effect sizes are specified as Cohen's d for interval scale variables and Cramér's V for nominal scale variables.*

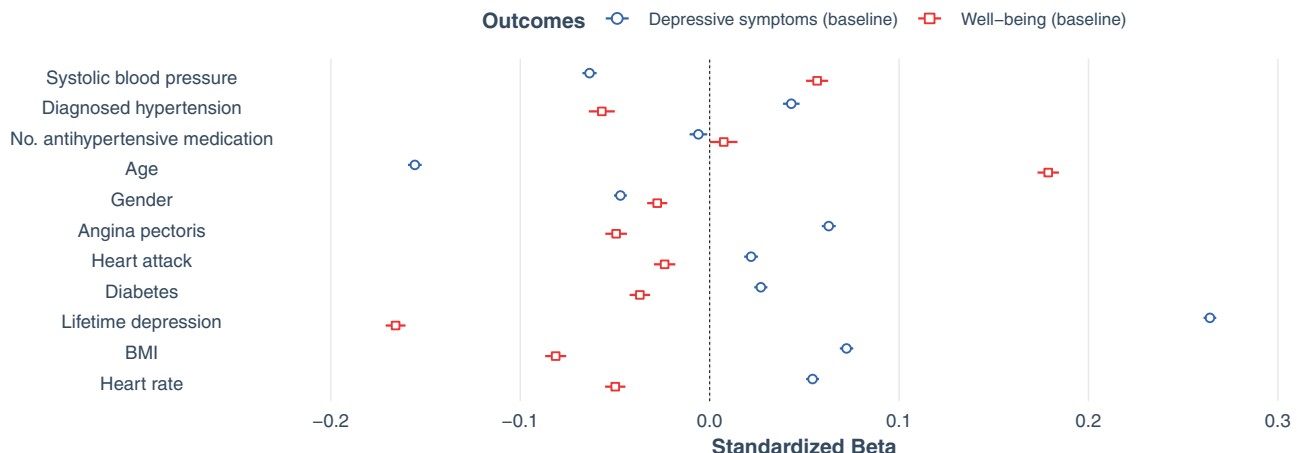

**Fig. 2 | Cross-sectional associations with mental health outcomes at initial assessment.** Forest plot shows standardized beta estimates and 95% confidence intervals for predictors of interest (systolic blood pressure, diagnosed hypertension (HTN), and number of antihypertensives) as well as covariates. There were

$N = 303,771$ participants with data for current depressive symptoms and $N = 129,876$ participants with data for well-being (after exclusion of missing values). Source data are provided as a Source Data file.

conducted on complete-case datasets. To account for bias due to missing data, we imputed data and conducted sensitivity analyses on the entire sample, which are reported below and in Supplementary Table 5. The results from imputed datasets indicated no sign of bias due to missing data and replicated our results from complete-case analyses (Supplementary Table 5).

**Outcome data**
At initial assessment, self-reported current depressive symptoms resulted in a mean score of 1.40 (SD = 0.53) and well-being in a mean score of 4.46 (SD = 0.58). At follow-up, depressive symptoms were reported with mean score of 1.30 (SD = 0.44) and well-being with a mean score of 4.63 (SD = 0.54).

Among participants who completed both initial and follow-up assessments ($N = 47,933$), correlation analyses revealed positive associations between baseline and follow-up measures for both depressive symptoms (Pearson $r = 0.529$, $p < 0.001$) and well-being (Pearson $r = 0.675$, $p < 0.001$), respectively. Thus, test-retest reliability of mood assessments was high, indicating stability over assessment timepoints.

**Confirmatory analysis I: Cross-sectional associations of systolic blood pressure, diagnosed hypertension, and antihypertensive medication intake with depressive symptoms and well-being**
For the multiple linear regression testing cross-sectional associations of SBP, HTN, and number of antihypertensive medications with depressive symptoms 303,771 participants could be included in the model, while 129,876 could be included for the multiple linear regression model with well-being as outcome. At baseline (Fig. 2), results of multiple linear regression models showed that SBP was negatively related to depressive symptoms ($\beta = -0.063$; 95% CI [−0.067, −0.060]; $p < 0.001$) and positively associated with well-being ($\beta = 0.057$; 95% CI [0.051, 0.063]; $p < 0.001$). Inversely, HTN was related to more depressive symptoms ($\beta = 0.043$; 95% CI [0.039, 0.047]; $p < 0.001$) and poorer well-being ($\beta = -0.057$; 95% CI [−0.064, −0.050]; $P < 0.001$). Restricting the analyses to participants with HTN only ($N = 107,192$), yielded similar associations of higher SBP with fewer depressive symptoms ($\beta = -0.054$; 95% CI [−0.060, −0.048]; $p < 0.001$) and greater well-being ($N = 45,319$), respectively ($\beta = 0.041$; 95% CI [0.032, 0.050]; $p < 0.001$). Furthermore, our analyses yielded a small negative relationship between the number of antihypertensive medications taken and depressive symptoms ($\beta = -0.006$; 95% CI [−0.011, −0.001]; $p = 0.012$), and a positive trend with well-being ($\beta = 0.007$;

95% CI [0.000, 0.015]; $p = 0.046$). In the analyses of participants with HTN, higher numbers of antihypertensive medications were similarly associated with fewer depressive symptoms ($\beta = -0.015$; 95% CI [−0.022, −0.009]; $p < 0.001$) and with greater well-being ($\beta = 0.010$; 95% CI [0.000, 0.020]; $p = 0.043$).

Overall, the model for depressive symptoms including all participants yielded a model fit of adj. $R^2 = 0.129$, which indicated an increase in model fit of $\Delta$adj. $R^2 = 0.004$ from a null model consisting only of covariates (well-being adj. $R^2 = 0.088$, $\Delta$adj. $R^2 = 0.004$; HTN only: depressive symptoms $\Delta$adj. $R^2 = 0.003$, well-being $\Delta$adj. $R^2 = 0.002$).

**Confirmatory analysis II: Longitudinal associations of systolic blood pressure, diagnosed hypertension, and antihypertensive medication intake with depressive symptoms and well-being**
For the multiple linear regression testing longitudinal associations of SBP, HTN, and number of antihypertensive medications with depressive symptoms 28,021 participants could be included in the model, while 29,966 could be included for the multiple linear regression model with well-being as outcome. Longitudinally (Fig. 3), we found that higher SBP was related to fewer depressive symptoms at follow-up assessment ($\beta = -0.042$; 95% CI [−0.055, −0.029]; $p < 0.001$) and to higher follow-up well-being scores ($\beta = 0.033$; 95% CI [0.020, 0.046]; $p < 0.001$). Similar to the cross-sectional results, baseline HTN was associated with more depressive symptoms approximately 10 years later ($\beta = 0.029$; 95% CI [0.014, 0.044]; $p < 0.001$). HTN was also significantly related to lower well-being scores at follow-up ($\beta = -0.032$; 95% CI [−0.047, −0.017]; $p < 0.001$). Number of antihypertensive medications at baseline was not a significant predictor in any longitudinal model (depressive symptoms: $\beta = -0.006$; 95% CI [−0.022, 0.009]; $p = 0.418$; well-being: $\beta = -0.004$; 95% CI [−0.019, 0.011]; $p = 0.620$). The model fit for the prediction of depressive symptoms increased from a null model consisting only of covariates by $\Delta$adj. $R^2 = 0.001$ (adj. $R^2 = 0.092$; well-being adj. $R^2 = 0.055$, $\Delta$adj. $R^2 = 0.001$).

**Exploratory analysis I: Moderation of the blood pressure-mental health relationship by development of hypertension**
Next, we explored if the relationship between mental health and SBP was moderated by hypertension status at follow-up assessment. First, we excluded participants who were hypertensive at the initial

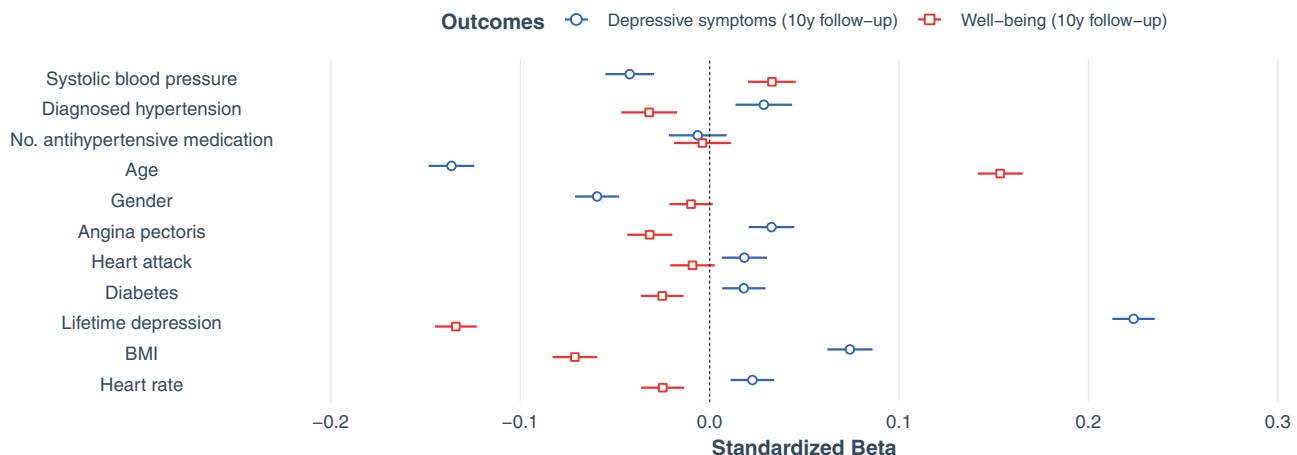

**Fig. 3 | Longitudinal associations with mental health outcomes at follow-up assessment.** Forest plot shows standardized beta estimates and 95% confidence intervals for predictors of interest at baseline (systolic blood pressure, diagnosed hypertension (HTN), and number of antihypertensives) as well as covariates. There were $N = 28,021$ participants with data for current depressive symptoms and $N = 29,966$ participants with data for well-being (after exclusion of missing values). Source data are provided as a Source Data file.

assessment (defined as HTN diagnosis or intake of antihypertensives). This resulted in 315,582 people who could be considered non-hypertensive at initial assessment and who were included in the analysis (of those, $N = 25,991$ had data available to define hypertension at follow-up). Across all participants, SBP increased from initial assessment to follow-up (Fig. 4A, mean increase = 3.820 mmHg; $t = 38.996$; degrees of freedom = 25,578; 95% CI [3.628, 4.012]; $p < 0.001$). People who later developed HTN already had higher SBP levels at initial assessment compared to people who stayed normotensive (Fig. 4B, HTN: mean baseline SBP (SD) = 148 (18.7) mmHg; no HTN: mean baseline SBP (SD) = 130 (15.3) mmHg). Notably, in unadjusted models, there were no significant group differences in mental health at initial assessment between people who developed HTN and those who stayed normotensive (Fig. 4C, HTN: mean depressive symptoms (SD) = 1.333 (0.5); no HTN: mean depressive symptoms (SD) = 1.327 (0.4); $t = -0.738$; degrees of freedom = 4585.9; 95% CI [−0.023, 0.010]; $p = 0.464$; Fig. 4D, HTN: mean well-being (SD) = 4.536 (0.6); no HTN: mean well-being (SD) = 4.547 (0.5); $t = 0.662$; degrees of freedom = 1583.8; 95% CI [−0.022, 0.044]; $p = 0.508$). However, in the fully adjusted regression model, we observed a main effect of later HTN on baseline mental health (Fig. 4C, depressive symptoms: $\beta = 0.060$; 95% CI [0.045, 0.076]; $p < 0.001$; Fig. 4D, well-being: $\beta = -0.043$; 95% CI [−0.068, −0.017]; $p < 0.001$), suggesting that when adjusting for SBP levels, people who later developed HTN had lower mood (i.e., more depressive symptoms and lower well-being) already at initial assessment compared to people without HTN. The regression model further yielded a significant interaction of SBP at initial assessment and HTN at follow-up with depressive symptoms at initial assessment, showing that the association between SBP and depressive symptoms was moderated by hypertension status at follow-up (Fig. 4E, $\beta = -0.014$; 95% CI [−0.026, −0.003]; $p = 0.015$). Thus, the slope of the relationship between higher SBP and fewer depressive symptoms was steeper in those participants who developed HTN approximately 10 years later. At follow-up, however, HTN status was not a significant moderator for the association of SBP and depressive symptoms ($\beta = -0.005$; 95% CI [−0.016, 0.007]; $p = 0.419$). Similarly, a trend of a moderation by HTN status was observed for the association between well-being and SBP at initial assessment, but this effect was not significant (Fig. 4F, $\beta = 0.017$; 95% CI [−0.001, 0.036]; $p = 0.070$). Neither was there a moderation by HTN status for the association between well-being at follow-up and SBP at follow-up ($\beta = 0.002$; 95% CI [−0.010, 0.013]; $p = 0.770$).

We replicated these findings using SBP > 140 mmHg as an additional criterion to define hypertension. This resulted in a sample where the two groups had SBP levels in the normotensive range at baseline. The results within this sample remained virtually unchanged and are reported in detail in the Supplementary Results.

## Exploratory analysis II: Associations with emotion-related brain function

To explore a central nervous representation of emotional reactivity, we assessed how hypertension status relates to emotional processing in the brain using fMRI activity assessed during the Hariri task. BOLD fMRI activity to emotional faces was lower in the amygdala (HTN = 1.134; no HTN = 1.249; $t = 9.797$; degrees of freedom = 15,129; 95% CI [0.091, 0.137]; $p < 0.001$) and in significant regions resulting from whole-brain analyses (HTN = 2.322; no HTN = 2.592; $t = 14.528$; degrees of freedom = 14,888; 95% CI [0.233, 0.306]; $p < 0.001$) in people with HTN compared to normotensives (Fig. 5 left panels). We also observed a negative association between lower BOLD reactivity to emotional faces and higher SBP across participants, suggesting a continuously negative effect of SBP on BOLD (amygdala: $\beta = -0.041$ 95% CI [−0.054, −0.028]; $p < 0.001$; whole-brain: $\beta = -0.032$ 95% CI [−0.045, −0.019]; $p < 0.001$, Fig. 5 middle panels). Additionally, an interaction of HTN and SBP suggested that the negative association between SBP and whole-brain BOLD activity was less pronounced in people with HTN (whole-brain: $\beta = 0.015$ 95% CI [0.003, 0.028]; $p = 0.014$; trend in amygdala: $\beta = 0.024$ 95% CI [−0.001, 0.024]; $p = 0.063$; Fig. 5 right panels). We repeated the same analysis using the blood pressure measurement at initial assessment which was acquired ~10 years prior to neuroimaging and thus not directly related to blood flow and volume dependent effects during BOLD fMRI. Interestingly, already the baseline SBP (i.e., prior to any HTN diagnosis) was negatively correlated with BOLD reactivity to emotional faces. In addition, there was an interaction of baseline SBP and later HTN on emotion-related brain activity, specifically in amygdala regions ($\beta = 0.017$; 95% CI [0.004, 0.030]; $p = 0.009$), suggesting that the negative correlation between SBP at baseline and emotion-related amygdala activity was dampened in participants who developed hypertension. The interaction between baseline SBP and BOLD activity was not significant in the whole-brain analysis ($\beta = 0.008$; 95% CI [−0.005, 0.021]; $p = 0.213$).

## Additional and sensitivity analyses

We performed several additional and sensitivity analyses to test the robustness of these results. Additional analyses included (i) using the

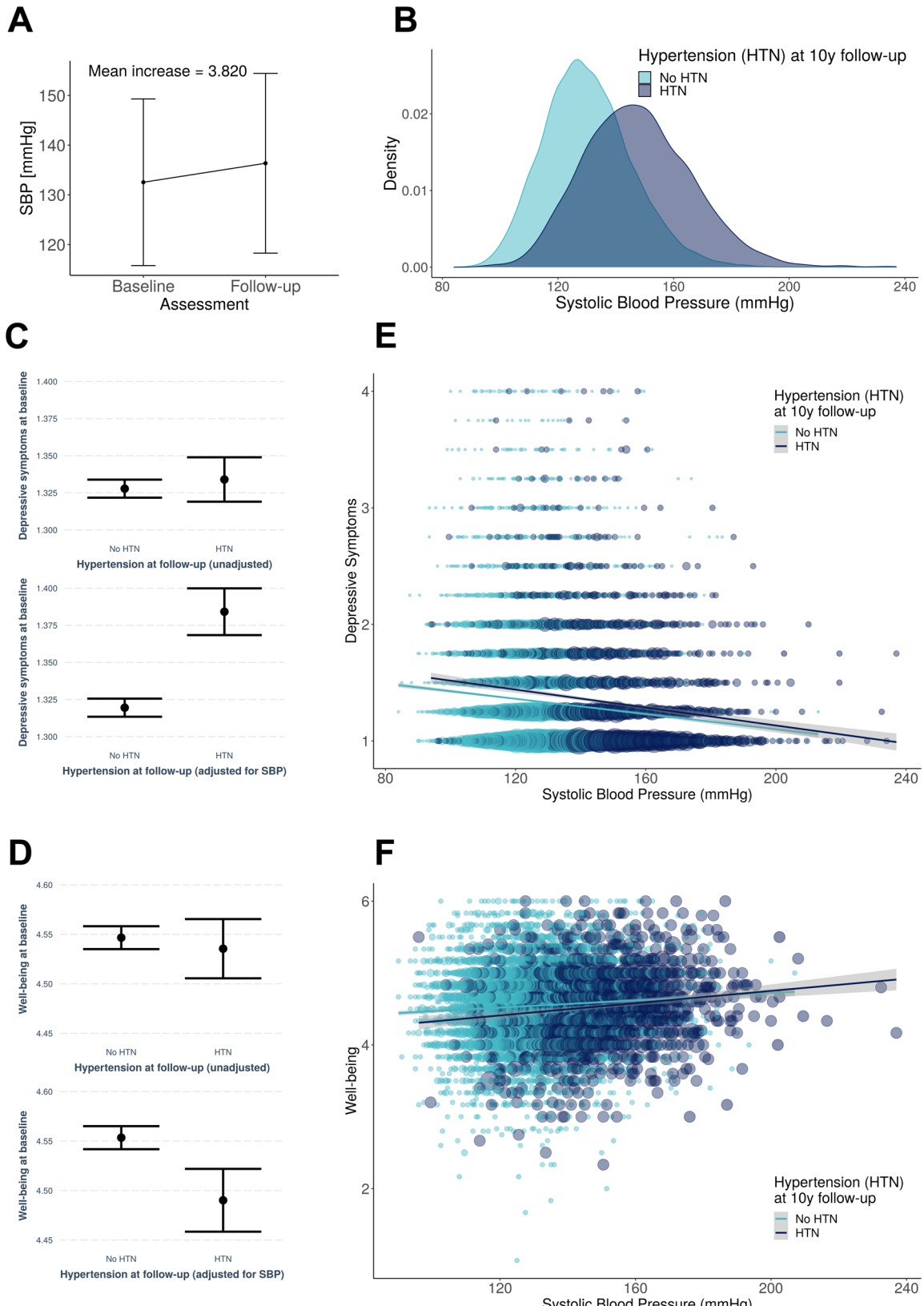

PHQ-9 questionnaire of the 5-year online mental health follow-up as a validated instrument to assess current depressive symptoms, (ii) cross-sectional analyses of mental health outcomes at both follow-up assessments, (iii) association of depressive symptoms within systolic blood pressure categories, (iv) additional relevant covariates, such as socioeconomic status, insomnia, racial/ethnic background, insomnia,

etc., (v) using hospital inpatient data for diagnoses of HTN and depression, and (vi) using SBP > 140 mmHg as an additional criterion to define HTN in the moderation analysis. Moreover, sensitivity ana-lyses were performed to test whether the above reported confirmatory results were dependent on (vii) the presence or absence of previous diagnosis of depression or any other severe disease that might affect

**Fig. 4 | Association between mental health and systolic blood pressure at initial assessment moderated by hypertension status at follow-up (i.e., approximately 10 years later). A** Across all participants (excluding those with HTN and use of antihypertensives and missing data, $N = 25,579$), systolic blood pressure increased from baseline to follow-up. **B** People who developed hypertension until follow-up (i.e., received a hypertension diagnosis or started taking anti-hypertensives) until the follow-up assessment (dark blue colour) already showed higher systolic blood pressure levels at initial assessment, despite not having been diagnosed at this timepoint, yet. **C** No significant difference between groups in depressive symptoms at baseline (top, $N = 24,202$), but when controlling for SBP, HTN developers showed more depressive symptoms (bottom, $N = 24,202$). **D** No

significant difference between groups in well-being at baseline (top, $N = 9,444$), but when controlling for SBP, HTN developers showed lower well-being (bottom, $N = 9444$). **E** Participants who developed hypertension (dark blue colour) had a steeper negative slope for the relationship between blood pressure and depressive symptoms than those participants who stayed non-hypertensive (light blue colour). **F** Similar trend for well-being in the expected opposite direction. Data in **A** are presented as mean values +/− SD. Data in **C** and **D** are presented as mean values +/− SEM. Data in **E** and **F** show individual data points with size of points representing density and regression lines with 95% confidence intervals. Source data are provided as a Source Data file.

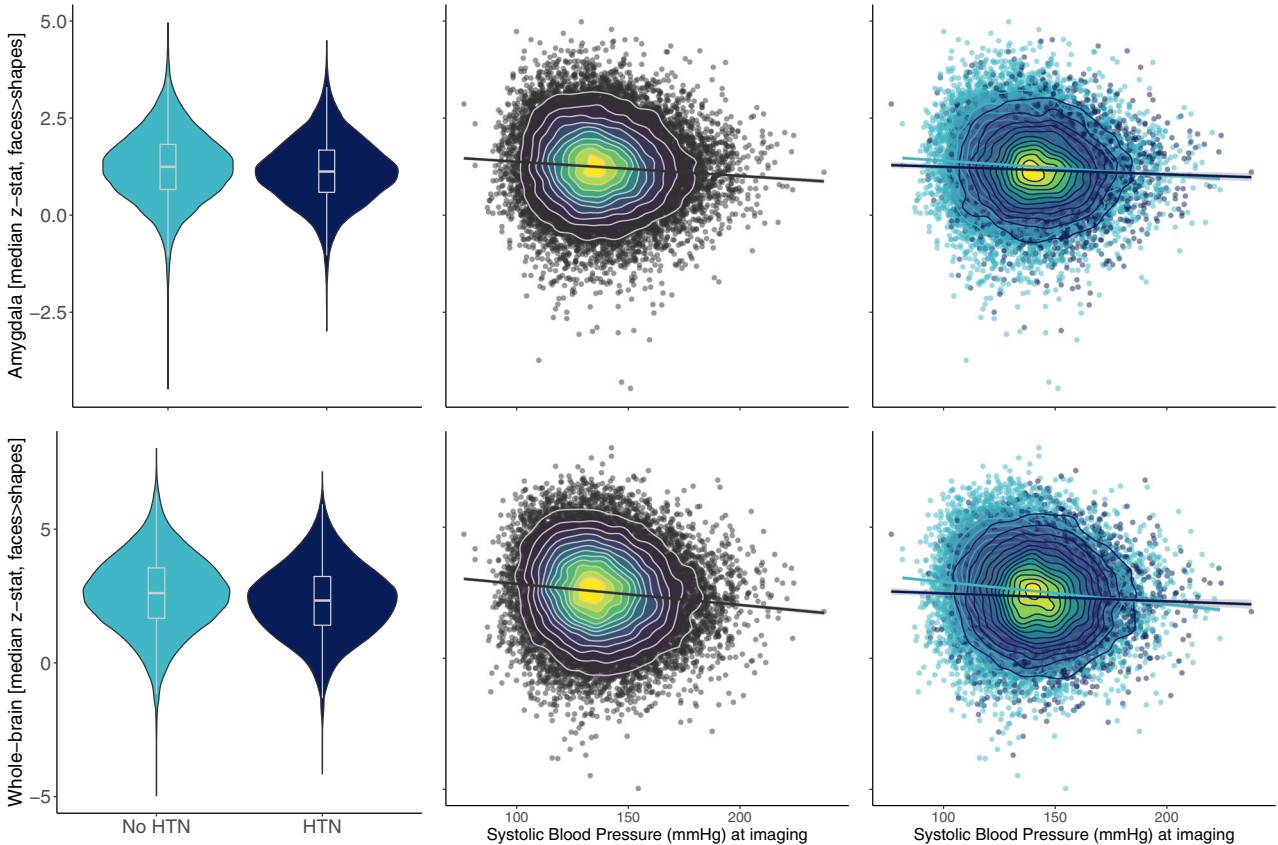

**Fig. 5 | Association between systolic blood pressure and emotion-related brain function at follow-up.** Left column (violin plots with boxplots presenting the median, lower and upper hinges corresponding to the 25th and 75th percentiles and lower/upper whiskers extending from the hinge to the smallest/largest value no further than 1.5 * IQR from the hinge) shows group differences in Hariri task activity by hypertension status (HTN) at follow-up/imaging visit ($N = 26,697$): BOLD fMRI activity to emotional faces was lower in the amygdala (HTN $z = 1.134$; no HTN $z = 1.249$; two-sided $t = 9.797$; degrees of freedom = 15,129; 95% CI [0.091, 0.137]; $p < 0.001$) and in significant regions resulting from whole-brain analyses (HTN $z = 2.322$; no HTN $z = 2.592$; two-sided $t = 14.528$; degrees of freedom = 14,888;

95% CI [0.233, 0.306]; $p < 0.001$) in people with HTN compared to normotensives. Middle column shows negative correlations between blood pressure and BOLD fMRI activation in the faces>shapes contrast of the Hariri task in amygdala mask and whole-brain mask. The colour gradient represents the density of data points. Right column shows the same, but grouped by HTN at follow-up. Dark blue colours represent people who became hypertensive from baseline to follow-up. Light blue colours represent participants who stayed normotensive. The negative correlation between systolic blood pressure and emotion-related activity was flattened in participants who had developed hypertension. Source data are provided as a Source Data file.

BP (list of diseases in Supplementary Table 2); (viii) the intake of antidepressants or any other medication intake; (ix) a specific effect of certain antidepressant or antihypertensive drug classes. Finally, we (x) performed multiple imputation of missing data, (xi) assessed potential survival bias, (xii) explored potential unmeasured confounding effects with E-values, and (xiii) applied joint modelling of time-varying effects of SBP and HTN using Linear Mixed Effects Models.

The results of all additional and sensitivity analyses are reported below and in the Supplementary Materials. In sum, the results were overall robust and consistent: independent of any potential confounders including medication intake, there was an effect of higher

SBP on fewer depressive symptoms and higher well-being, as well as a negative effect of HTN diagnosis on mental health.

**5-year online mental health follow-up results using the PHQ-9.** Longitudinally, we found that baseline HTN was associated with more depressive symptoms assessed with the PHQ-9 approximately 6 years after the initial assessment at the online mental health follow-up ($\beta = 0.033$; 95% CI [0.025, 0.041]; $p < 0.001$). Similar to the cross-sectional results, higher SBP was related to fewer depressive symptoms at the 6-year-follow-up assessment (PHQ-9: $\beta = -0.049$; 95% CI [−0.056, −0.042]; $p < 0.001$). Number of antihypertensive medications

at baseline was not significantly associated with the PHQ-9 depression score. The results from the PHQ-9 highly resemble the results using the depression score reported in the main manuscript (Supplementary Fig 4).

**Cross-sectional association of mental health outcomes at both follow-up assessments.** We further analysed cross-sectional models at both follow-up time points. Despite smaller sample sizes, resulting in larger 95% confidence intervals, the results yielded robust, significant estimates for SBP in line with our previous analyses (Supplementary Fig 5). Estimates for HTN diagnosis showed that the direction of effects was positive for depressive symptoms ($\beta = 0.007$; 95% CI [−0.017, 0.030]; $p = 0.579$) and negative for well-being ($\beta = −0.016$; 95% CI [−0.038, 0.007]; $p = 0.183$) at 10-year follow-up, consistent with the main findings, yet, results were not significant. For the number of antihypertensive medications, we observed significant associations with all mental health outcomes, suggesting that higher hypertension burden, indicated by a higher number of prescribed antihypertensives, negatively relates to mood.

**Association of depressive symptoms within systolic blood pressure categories.** We additionally evaluated how the association between SBP and depression presents in varying quantiles along the range of SBP levels, similar to previous studies[33–35].

We divided the data into 7 bins following the commonly used categories for diagnostic SBP classification ("<90" [Reference Category], "90-120", "120-130", "130-140", "140-160", "160-180", ">180" mmHg). Next, we computed the cross-sectional association between SBP bins and depressive symptoms at baseline while adjusting for the same covariates as in the main manuscript.

Interestingly, the results indicate fewer depressive symptoms for each unit increment in bins of SBP (Supplementary Fig 6), suggesting that SBP positively affects mood symptoms at greater rates in higher ranges of SBP levels until it decreases again in ranges of hypertensive crises (>180 mmHg). The estimate for SBP between 90-120 mmHg was not significantly different from the reference category (estimate = −0.070, 95% CI [−0.167 0.027], $p = 0.156$), which supports our findings that the negative association between SBP and mood might be particularly salient for individuals at risk of high blood pressure.

**Modelling of additional relevant variables.** Further influencing factors have been previously shown to be important effect modifiers for associations with mental health and/or hypertension, such as insomnia[36], socioeconomic status and education[37,38], as well as race and ethnic background[39]. To assess the robustness of our cross-sectional results to these effects, we analysed an extended version of our cross-sectional model adjusting additionally for insomnia, household income, educational attainment, and racial/ethnic background. Racial/ethnic background was recoded as a dichotomous variable (White / People of Colour [including people who identified as Asian or Asian British, Black or Black British, Chinese, Mixed, or Other]) as the vast majority of participants self-identified as white. We also included the UK Biobank assessment centre to model that data acquired at the same assessment centre might be correlated. Among these additionally included covariates (Supplementary Fig 7), insomnia had the largest effect (i.e., standardized beta) on both depressive symptoms ($\beta = 0.224$; 95% CI [0.221, 0.228]; $p < 0.001$) and well-being ($\beta = −0.172$; 95% CI [−0.177, −0.166]; $p < 0.001$). While all other additionally included covariates also had some effect on mental health, the effects of our main variables of interest (SBP, HTN, No. of antihypertensives ) on both mental health outcomes remained virtually unchanged when controlling for all these confounders.

**Analyses with hospital records (HES).** To complement our results using self-report data with clinical assessments, we repeated the confirmatory analyses using hospital diagnoses for HTN and depression from the HES database. We used the secondary ICD-10 diagnoses codes a participant has had recorded across all their hospital inpatient records (Supplementary Table 1). For HES-diagnosed HTN, we included all occurrences of code I10 'Essential (primary) hypertension'. For HES-diagnosed depression, we included all occurrences within codes F32 'Depressive episode' and F33 'Recurrent depressive disorder' without psychotic symptoms. The occurrences were summed per participant and coded as 0 or 1 for the absence or presence of HTN or depression, respectively.

Among the baseline sample ($N = 502,494$), a total of 340,896 individuals had HES data available. HTN was diagnosed in 112,554 (33%) of HES cases and depression in 18,274 (5%) of HES cases, which shows that HES records may underestimate the prevalence of these conditions in the general population[40,41]. Agreement between self-reported conditions and HES diagnoses was assessed using Cohen's kappa. In line with previous research[42], we observed moderate agreement for HTN (kappa = 0.583) and fair agreement for depression (kappa = 0.324).

Next, we used HES-diagnosed HTN to repeat our analysis testing the cross-sectional relationship between depressive symptoms/well-being and SBP, HTN and antihypertensive medication. Including HES-diagnosed HTN instead of self-reported HTN replicated our main results (Supplementary Fig 8). We again observed that SBP was negatively related to depressive symptoms ($\beta = −0.064$; 95% CI [−0.068, −0.060]; $p < 0.001$), whereas HTN was related to more depressive symptoms ($\beta = 0.069$; 95% CI [0.065, 0.073]; $p < 0.001$) and there was a negative relationship between the number of antihypertensive medications taken and depressive symptoms ($\beta = −0.011$; 95% CI [−0.015, −0.006]; $p < 0.001$). The results yielded larger effect size estimates (i.e., standardized betas) for all these predictors as compared to the analysis with self-reported HTN (Supplementary Fig 8). Inversely, SBP was positively associated with well-being ($\beta = 0.054$; 95% CI [0.048, 0.059]; $p < 0.001$), yet the standardized beta coefficient was slightly lower than in the analysis using self-reported HTN. HES HTN related to poorer well-being ($\beta = −0.068$; 95% CI [−0.074, −0.061]; $p < 0.001$) with a larger standardized beta coefficient than in the analysis using self-reported HTN. Finally, the relation between antihypertensives and well-being was also not significant in the model including HES HTN ($\beta = 0.004$; 95% CI [−0.002, 0.011]; $p = 0.193$). Overall, the models including HES HTN explained slightly more variance, as reflected in greater adjusted $R^2$-values compared to the models using self-report HTN (depressive symptoms: adj. $R^2 = 0.131$; well-being: adj. $R^2 = 0.089$).

Finally, we included an additional analysis testing the relationship of SBP, self-reported HTN and antihypertensive medication with occurrence of HES-diagnosed depression ($N = 340,900$, Supplementary Fig 9). This analysis yielded converging effects with our findings based on self-reports, showing that higher SBP was associated with a lower occurrence of HES-recorded depression ($\beta = −0.040$; 95% CI [−0.043, −0.036]; $p < 0.001$), whereas a HTN diagnosis showed a small effect in the opposite direction ($\beta = 0.009$; 95% CI [0.005, 0.014]; $p < 0.001$). There was also a small positive association between the number of antihypertensives taken and the occurrence of depression ($\beta = 0.008$; 95% CI [0.004, 0.013]; $p = 0.001$). The overall model fit was low (adj. $R^2 = 0.010$).

**Moderation analysis of BP-mental health relationship with SBP as additional criterion to define hypertension status**
In the moderation analysis (Fig. 4), we observed that among people who were regarded as "normotensive" based on the two criteria "lack of a previous HTN diagnosis" and "no intake of antihypertensive medication", there were some individuals with baseline systolic blood pressure levels of 140 mmHg or higher. We therefore refined our criteria to thoroughly detect all hypertensive participants at both initial assessment (for conservative exclusion) and at follow-up (to clearly

define who will develop hypertension). We thus repeated the moderation analysis while considering SBP > 140 mmHg as an additional criterion (next to HTN diagnosis and antihypertensive medication) for the definition of hypertension. Thus, we excluded all participants who were hypertensive at the initial assessment (defined as SBP > 140 mmHg, HTN diagnosis or intake of antihypertensives), leaving us with a sample of $N = 17,879$ participants with data at baseline and follow-up. In this sample of non-hypertensive participants, mean baseline SBP was 124 mmHg (SD = 9.92). Among these, those who stayed normotensive until follow-up had a mean SBP of 122 mmHg (SD = 10.0) and those who developed HTN had a mean SBP of 129 mmHg (SD = 7.76). Across all non-hypertensive participants, SBP increased significantly from initial assessment to follow-up (mean increase = 6.885 mmHg; $t = 67.432$; degrees of freedom = 17,878; 95% CI [6.686, 7.086]; $p < 0.001$). Within this sample, we replicated the results obtained in the larger group: In unadjusted models, there were no significant group differences in mental health at initial assessment between people who developed HTN and those who stayed normotensive (HTN: mean depressive symptoms = 1.347; no HTN: mean depressive symptoms = 1.358; $t = 1.346$; degrees of freedom = 9608; 95% CI [−0.005, 0.026]; $p = 0.178$; HTN: mean well-being = 4.525; no HTN: mean well-being = 4.523; $t = −0.109$; degrees of freedom = 2888.2; 95% CI [−0.033, 0.029]; $p = 0.913$). However, in the fully adjusted regression model (e.g., adjusting for baseline blood pressure), we again observed a main effect of later HTN on baseline mental health (depressive symptoms: $\beta = 0.038$; 95% CI [0.021, 0.056]; $p < 0.001$; well-being: $\beta = −0.029$; 95% CI [−0.058, 0.001]; $p = 0.056$), suggesting that when adjusting for SBP levels, people who later developed HTN reported more depressive symptoms already at initial assessment compared to people without HTN. We also replicated that the negative association between depressive symptoms and SBP at initial assessment was moderated by HTN at follow-up ($\beta = −0.021$; 95% CI [−0.039, −0.004]; $p = 0.019$). The moderation was however not significant for depressive symptoms at follow-up. Neither did we observe any significant moderation effects of developing HTN for well-being at any of the assessments. In sum, these results corroborate our findings that people who developed HTN throughout the course of the study might require higher SBP levels to sustain the same mental health outcomes as normotensives and that the negative relationship between mental health and blood pressure was accentuated in people developing HTN several years before HTN manifested.

### Inclusion/exclusion of previous depression and other BP-altering diseases

We also tested if any effects of SBP, HTN and antihypertensive intake on current depressive symptoms and well-being depended on the presence or absence of previous diagnosis of depression or any other severe neurological, systemic, or psychiatric diseases that might affect BP (e.g., stroke, CVD, renal diseases, schizophrenia; list of diseases in Supplementary Table 2). We thus performed the same multiple linear regression models described above separately for groups of participants with or without any of these diseases.

Conducting the analyses separately in groups of participants who either have had no or any previous diagnosis of depression yielded similar cross-sectional results as in the total sample (all $p < 0.001$, Supplementary Table 3). As described in Supplementary Table 3, in people both with or without depression diagnosis, SBP was negatively associated with depressive symptoms and positively with well-being. Inversely, in both sub-groups HTN was positively related with depressive symptoms and negatively with well-being. Higher intake of antihypertensives was related to fewer depressive symptoms only in people without previous depression.

The results were also robust in sub-samples of people without or with diseases which often co-occur with changes in blood pressure, such as cardiovascular or renal diseases (Supplementary Table 3). Only

number of antihypertensives yielded diverging results between these sub-groups: While people without disease diagnoses, showed a negative association between number of antihypertensives and depressive symptoms and a positive one with well-being scores, higher intake of antihypertensives was related to more depressive symptoms and lower well-being in people with any diseases (Supplementary Table 3).

### Medication effects

A major characteristic of the UK Biobank sample is the large number of people taking any kind of medication ($N = 363,967$). Among the most frequently used medications were pain killers (e.g., paracetamol, $N = 93,619$), antihypertensives (e.g., bendroflumethiazide, $N = 28,000$) and antidepressants ($N = 37,699$). We therefore explored potential effects of these medications on the cross-sectional relationship between blood pressure and mental health at baseline. Again, in separate multiple regression models of groups of participants taking or not taking these medications, we added SBP, HTN and number of antihypertensive medications simultaneously as predictors of depressive symptoms and well-being and corrected the models with the same covariates as described in the confirmatory analyses above.

**Antidepressants.** Sub-group analyses of participants taking or not taking antidepressant medication, showed similar associations between SBP, HTN and mental health as analyses on the total sample (Supplementary Table 4). However, when considering only participants not taking antidepressants, higher intake of antihypertensives was associated with fewer depressive symptoms ($\beta = −0.013$; 95% CI [−0.018, −0.008]; $p < 0.001$) and greater well-being ($\beta = 0.009$; 95% CI [0.002, 0.017]; $p = 0.018$).

### Other medications

We also compared the blood pressure-mental health relationship between participants taking any sort of medications and those who took none (Supplementary Table 4). Again, we found a negative effect of SBP and a positive effect of HTN on depressive symptoms (and the inverse pattern for well-being) in both sub-groups (Supplementary Table 4). Despite the smaller sample sizes, standardized beta estimates were slightly larger for the sub-group of participants who did not take any medications, compared to the total sample (Supplementary Table 4 and Fig. 2).

### Specificity of antidepressant classes

In addition, we tested whether there was a specific effect of antidepressant or antihypertensive drug classes on the relationship of blood pressure and mental health (see Supplementary Fig 2 and 3 for list of medication classes). For these analyses, we categorised antidepressants and antihypertensives according to their mechanisms of action. In separate multiple regression models, we added SBP, HTN and a categorical variable coding for the medication classes simultaneously as predictors of depressive symptoms and well-being and corrected the models with the same covariates as described in the confirmatory analyses above.

With regards to depressive symptoms, we found negative associations with tricyclic antidepressants (estimate = −0.296; 95% CI [−0.558, −0.033]; $p = 0.027$) and negative associations with unselective reuptake inhibitors (estimate = −0.354; 95% CI [−0.623, −0.084]; $p = 0.010$). Associations between well-being and antidepressant classes were not significant (all $p > 0.05$).

### Specificity of antihypertensive classes

We did not observe any significant effects of antihypertensive classes on depressive symptoms (all $P > 0.05$). Well-being was negatively associated with aldosterone antagonists (estimate = −0.261; 95% CI [−0.502, −0.020]; $p = 0.034$) and positively with the intake of a

combination of beta and alpha1 blockers (estimate = 0.347; 95% CI [0.029, 0.664]; $p = 0.032$).

## Multiple Imputation of missing data

To account for bias due to missing data, we imputed data and conducted sensitivity analyses on the entire sample. Analyses reported above were based on the complete-case data with listwise exclusion of missing values, and the cross-sectional models at baseline assessment were repeated in imputed data for comparison. Data were imputed for individuals without values for predictor, outcome and/or covariates using multiple imputation. Tabulation of missing data showed percentages of missing data below 10% for predictor and outcome variables (Table 1), except for well-being variables which were introduced at a later timepoint during the baseline assessment (see notes in related data field, e.g. https://biobank.ctsu.ox.ac.uk/crystal/field.cgi?id= 4526). Most missing data occurred in covariates for diagnoses of angina, heart attack and lifetime depression ($N = 127,467$; 24%). To impute datasets, we created 20 imputations using multiple imputation by chained equations with the R package *MICE* (version 3.14.0)[43,44]. Linear regression was performed on each imputed dataset and estimates of the results were then averaged across the imputed datasets in accordance with Rubin's rules[45]. The results from imputed datasets for the cross-sectional association between mental health and SBP/HTN/ number of antihypertensives indicated no sign of bias due to missing data and replicated our results from complete-case analyses (Supplementary Table 5).

## Assessment of survival bias

To assess potential survival bias in our sample, we used UK Biobank's death records and observed that 20,442 participants died between the two timepoints of our study, i.e., between the baseline visit and the imaging follow-up (~ 10 years later). The non-surviving sample was older, had higher blood pressure and reported more diagnoses than the total sample (Supplementary Table 6). To assess if there was any survival bias on our results, we investigated our covariate-adjusted cross-sectional model in the non-surviving sample ($N = 13,277$ for depressive symptoms as outcome; $N = 4,865$ for well-being as outcome). We again observed that SBP was negatively related to depressive symptoms ($\beta = -0.073$; 95% CI [$-0.090$, $-0.056$]; $p < 0.001$), whereas HTN was related to more depressive symptoms ($\beta = 0.047$; 95% CI [0.028, 0.067]; $p < 0.001$). There was no significant relationship between the number of antihypertensive medications taken and depressive symptoms ($\beta = -0.004$; 95% CI [$-0.026$, 0.019]; $p = 0.749$). Inversely, SBP was positively associated with well-being ($\beta = 0.063$; 95% CI [0.035, 0.092]; $p < 0.001$). The relationship of well-being with HTN ($\beta = -0.016$; 95% CI [$-0.049$, 0.017]; $p = 0.350$) was not significant, while well-being was negatively associated with number of antihypertensives ($\beta = -0.040$; 95% CI [$-0.078$, $-0.003$]; $p = 0.036$). The results yielded larger effect size estimates (i.e., standardized betas) for SBP as compared to the analysis in the surviving sample, which could indicate that the effect of SBP on mental health might be particularly relevant for populations at risk with very high SBP levels. Overall, the models in the non-surviving sample explained slightly more variance, as reflected in greater adjusted $R^2$-values compared to the models in the surviving sample (depressive symptoms: adj. $R^2 = 0.132$; well-being: adj. $R^2 = 0.098$).

## Exploration of unmeasured confounding bias with the E-value

To also explore the effect of potential unmeasured confounders on the results, we calculated E-values using the *EValue* package in R (version 4.1.3)[46]. E-values indicate the minimum strength of association, on the risk ratio scale, that an unmeasured confounder would need to have with both the exposure and outcome to explain away an exposure–outcome association, above and beyond the measured covariates. For continuous exposure variables, such as SBP in our study, the function uses a dichotomisation in the exposure between hypothetical groups of participants per one-unit increase. For the continuous outcome, such as depressive symptoms and well-being in our study, the function uses an effect-size conversion to approximately convert the mean difference between the exposure groups to the odds ratio that would arise from dichotomising the continuous outcome[46]. As a result, we observed for the association of SBP with depressive symptoms values of $E = 1.32$ and with well-being of $E = 1.30$. For the association of HTN with depressive symptoms, we observed $E = 1.25$ and with well-being $E = 1.30$. The interpretation of the E-value depends on the context regarding the measured confounders. Given that we have included several plausible confounders in our models, the resulting E-values give further support for the robustness of our results, as an unmeasured confounder would need to be associated with outcome and exposure by a risk ratio of around 1.3-fold each, above and beyond the measured confounders. However, we also note that while E-values have been introduced as robustness analyses for studies aiming to estimate causal effects[46], we consider them an additional sensitivity measure for robustness. Potential causal effects should more directly be investigated in future studies with experimental designs or observational designs with dedicated causal identification strategies (e.g., directed acyclic graphs).

## Joint time-varying analysis using Linear Mixed Effects Models

To directly model time-varying effects of SBP and HTN in a joint model that accounts for the unbalanced design in the follow-up measures, we explored the longitudinal association with mental health in a Linear Mixed Effects Model framework using R's *lmer* function (package *lme4*, version 1.1-23). Separate models were set up for depressive symptoms and well-being as outcomes, respectively. The full models further included a random intercept per participant, the same covariates of no interest as in the main analyses, fixed effects for SBP, HTN and number of antihypertensive medications, as well as three two-way interactions: SBP*HTN, SBP*time point and HTN*time point. Time point was coded as a dummy variable for "initial assessment" and "imaging follow-up". Each full model was compared to a null model that was identical but did not include the interactions. The difference between the full and the null model was tested using the *anova* function and setting the argument *test* to "chisq" to perform a Chi-squared test. Chi-squared values and p-values for each interaction effect were derived by dropping interactions iteratively (reduced models). Non-significant interactions were dropped from the full model to yield a less complex reduced model. For depressive symptoms, the full-null-model comparison was significant ($X^2(3) = 10.67$, $p = 0.014$). Iterative model reductions revealed that the interaction of SBP*time point was significant ($X^2(1) = 10.55$, $p = 0.001$), while HTN*timepoint ($X^2(1) = 1.78$, $p = 0.182$) and HTN*SBP ($X^2(1) = 0.11$, $p = 0.974$) were not significant. The significant interaction of SBP and time point indicates that the relationship between SBP and depressive symptoms changes between assessments. Evaluation of the fixed effects revealed that the interaction was driven by a steeper negative slope at baseline than at follow-up. The stronger negative relationship between SBP and depressive symptoms at baseline compared to follow-up supports our observation from multiple linear regression analyses showing that the relationship was stronger for the cross-sectional analysis at baseline than for the longitudinal analysis including follow-up outcomes. As expected, HTN diagnosis showed no significant interaction with time point, but a significant positive main effect ($X^2(1) = 358.69$, $p < 0.001$) indicating that the effect of HTN diagnosis on depressive symptoms was stable across assessments. No significant difference between individuals with and without HTN was found regarding the relationship between SBP and depressive symptoms across both time points (i.e., no significant SBP*HTN interaction). This observation is in line with the results from the moderation analysis, which showed a moderating effect of impending HTN on the relationship between SBP and

depressive symptoms at baseline assessment, but not at follow-up. For well-being, the full-null-model comparison was not significant ($X^2(3)$ = 5.39, $p = 0.145$). While we did not find significant interaction effects on well-being – suggesting that the main effects remained constant over time – the positive effect of SBP and negative effect of HTN on well-being reported in the main manuscript was corroborated by significant main effects of these variables (HTN: $X^2(1) = 256.54$, $p < 0.001$, SBP: $X^2(1) = 352.86$, $p < 0.001$). With these additional mixed model analyses, we were able to include all available data including participants with missing data; confirming the findings from multiple linear regression models reported in the main manuscript.

## Discussion

In this study, we confirmed two seemingly contradictory associations of high blood pressure with mental health: (i) Higher SBP was associated with fewer depressive symptoms and greater well-being at the initial exam as well as at the 5-year mental health online follow-up and the 10-year follow-up including imaging, whereas (ii) the presence of a HTN diagnosis was associated with more depressive symptoms and lower well-being. Given the well-powered cohort of the UK Biobank, we were able to perform sensitivity analyses which confirmed that the observed associations were robust to biases of medications (e.g., antihypertensives, antidepressants), chronic illness (e.g., CVD, clinical depression), survival, social factors and unmeasured confounds. Strikingly, we furthermore found that (iii) already at the initial exam, among normotensive individuals, mental health was negatively affected (more depressive symptoms, lower well-being) by the *later* HTN status at 10y-follow-up. Also, (iv) the relationship between blood pressure and mental health – at the initial exam – was moderated by later HTN, such that the negative association between blood pressure and depressive symptoms was stronger in those participants who later developed HTN. Finally, (v) our results from task-based functional brain imaging provide further support for an impact of blood pressure and HTN on central processing of emotions: We demonstrate a negative relationship between SBP – at baseline (~10 years prior to fMRI) and at the time point of imaging – and the BOLD fMRI response to aversive emotional stimuli. This relationship was again moderated by HTN status at follow-up, such that people who developed HTN showed overall lower responses to aversive stimuli and a flattened (negative) relationship between SBP and brain responses. Taken together, our results support the notion that the interrelation between blood pressure and mental health may be involved in the development of high blood pressure with potential implications for developing new preventive and therapeutic approaches for essential hypertension.

We preregistered and confirmed a conceptional replication that higher SBP related to better mood ratings: Our findings are consistent with studies reporting positive effects of elevated blood pressure on mental health, including decreased depressive symptoms[16–18], better quality of life[12,18], and reduced self-reported stress[13–15]. We extend these findings, which were based on either cross-sectional designs or shorter follow-up periods (1-5 years), by demonstrating that baseline SBP remained a significant predictor of mental health up to 10 years later. The positive effects of high blood pressure on mood might be related to research findings showing that higher blood pressure diminishes emotional experience in experimental manipulations[47–49]. It is also well established that elevated blood pressure robustly reduces the perception of physical pain[21,25,50,51], but also social pain[52]. It has therefore been hypothesised that elevated blood pressure relates to a generalised attenuation of emotional valence processing[25,49]. Our fMRI findings are consistent with this notion, suggesting an impact of blood pressure on the processing of (negative) emotional faces, even at a follow-up after 10 years. Importantly, while affective attenuation might relate to coping mechanisms to lift mood in stressful situations, it

could reinforce staying in potentially harmful circumstances over prolonged periods of time. This may lead to further blood pressure increases and eventually to HTN (see below)[20,25,27–29].

Given these consistent findings of a positive association between blood pressure and mood, it seems paradoxical that we found a different pattern for diagnosed HTN; albeit again in line with previous studies in which the presence of vascular risk factors or manifest CVD has been associated with increased depressive symptoms[53,54] and decreased well-being[55]. Several potential explanations have been put forward: Biological explanations build on well-known pathophysiological consequences of chronic blood pressure elevations (e.g., atherosclerosis, small vessel disease, etc.) leading to ischaemic brain damage indicated by white matter lesions, microinfarcts, and cerebral micro-haemorrhages[56]. White matter lesions, in particular, have been linked to the occurrence of depression with a vascular component[57,58]. Systemic mechanisms resulting from unfavourable metabolic alterations, which are common in people with HTN, and an unhealthy lifestyle (e.g., smoking, physical inactivity, unbalanced diet, etc.) have also been linked to depression via metabolic, immuno-inflammatory, and autonomic pathways (reviewed in[59]). Psychological explanations, on the other hand, emphasise that individuals receiving a diagnosis of HTN are often confronted with a sudden awareness of a chronic illness that requires medical attention and a change of lifestyle. The negative psychological consequences of such a labelling effect could underlie opposing effects of elevated blood pressure and HTN diagnosis on mental health[60,61].

Based on our data, we cannot exclude contributions of the factors discussed above, however, a major finding of our study (i.e., that an impact of HTN on mood is already present before the diagnosis), is difficult to reconcile with these explanations: We observed – as expected – that those participants who later developed HTN already had higher blood pressure at the initial visit than those who stayed normotensive. Yet, in the fully adjusted model, i.e., correcting for the baseline differences in blood pressure, the negative impact of (later) HTN on mental health was already significantly present *before* the HTN diagnosis. We found this in two analyses, one in which HTN was defined by previous HTN diagnosis and intake of medication, and one in which SBP > 140 mmHg was also used as criterion. In both analyses – when unadjusted for blood pressure – there were no significant differences in mental health at either visit, while the difference was highly significant when adjusted for blood pressure. Interestingly, we also noted in both analyses that the two groups differed regarding the overall mood-lifting effect of higher blood pressure at the initial visit, such that this effect was more pronounced in the group of those participants who developed HTN later. Thus, it seems that in the HTN-developing group, the relationship between blood pressure and mental health was both shifted in magnitude and had a different slope. This finding cannot be explained by the labelling effect and is also unlikely to be related to vascular damage, such as white matter lesions, as these occur only after long-lasting blood pressure elevations.

Obvious candidates for explaining effects of blood pressure on mental health are regulatory circuits linking arterial blood pressure to central processing in the brain. While the causal, and likely multifactorial, pathways between blood pressure and mental health are not fully understood, a shared mechanism between subjective experience, emotional processing and pain involves the regulatory baroreflex system. Baroreceptors, stretch sensitive receptors located in the aortic arch and the carotid artery sinus, are the peripheral sensors of blood pressure[62–64]. During each heartbeat, baroreceptors are activated during systole and become less active during diastole. They are known to relay phasic and tonic information about blood pressure via the vagal and glossopharyngeal nerves to brain stem nuclei which orchestrate adjustments of blood pressure and heart frequency via the parasympathetic and sympathetic nervous system[62–64]. Importantly, in addition to their role in adjusting blood pressure and heart frequency,

baroreceptor activation has also been shown to influence emotional and pain processing (reviewed by Suarez-Roca et al.[64]), thereby mediating behavioural and central effects of blood pressure modulations. Direct evidence comes from animal studies, in which for example, pain-relieving effects of blood pressure elevations were abolished by baroreceptor denervation[20,27] and from studies in humans in whom local baroreceptor stimulation modulated pain perception[21,24,25,27]. Further evidence has been provided by numerous studies showing different processing of pain, emotion, and sensory stimuli in systole versus diastole[22,23,26,65]. Importantly, it is also well established that the development of HTN is characterised by a progressive desensitisation of baroreceptors and altered sensory processing[27,66]. It, therefore, seems highly plausible that (relatively reduced) baroreceptor signalling might also underlie the observed altered relationship between blood pressure and mental health in hypertensive people. Our results furthermore indicate that the altered relationship between blood pressure and mental health may already be present years before the diagnosis of HTN. In a similar vein, our fMRI findings show an altered relationship between blood pressure and BOLD activation to negative facial expressions in people with HTN, consistent with adjusting central processing of emotions as a response to baroreceptor desensitisation. In sum, there is evidence that baroreceptor signalling can underlie the effect of higher blood pressure on mental health.

With regard to the development of blood pressure increases over the life course and eventually the pathophysiology of arterial HTN, our findings are consistent with the notion of feedback loops wherein arousing emotional stimuli and stressors elevate blood pressure, which in turn activates baroreceptor pathways that induce analgesia and decrease the perceived affective magnitude of a stressor[29]. While psychosocial stress is increasingly accepted as a risk factor for the development of hypertension[67], blood pressure adjustments – via baroreceptor signalling – may link stress to a rewarding mechanism decreasing perceived stress. Reinforcement by repeated stress exposure may eventually lead (or contribute) to increases in blood pressure and the development of essential HTN[20,25,27–29]. Our data further emphasise inter-individual differences: Those individuals who later developed HTN on average showed lower mental health scores when adjusting for SBP. In addition, our moderation analysis yielded that the development of HTN was associated with a stronger negative correlation between mental health and blood pressure at baseline, years before the HTN diagnosis. While the observed *inter*-individual differences cannot be readily interpreted as indicative of *intra*-individual mechanisms, it is nevertheless tempting to speculate that people at higher risk of developing HTN require higher blood pressure levels to sustain the same mental health outcomes. These individuals may find themselves on a relatively steeper trajectory towards HTN due to the stronger mood-lifting effect of blood pressure increases. Taken together we propose that (i) feedback loops between blood pressure and rewarding emotional processing during periods of stress may play a role in the pathophysiology of blood pressure increases and HTN and that (ii) alterations of these feedback loops characterised by a shifted blood pressure - mental health relationship may increase HTN risk in affected individuals. Based on our data, we cannot differentiate between potential reasons for the altered blood pressure-mental health relationship in the mostly middle-aged participants, which may include genetics, life-style factors, such as nutrition or the environment, previous exposure to acute and chronic stress, or other factors. Clearly, there is a need for prospective longitudinal studies clarifying this issue.

Beyond effects of blood pressure and diagnosed HTN, other factors such as medication and previous diseases can influence mental health and thus be confounders in our analyses. For example, Hermann-Lingen et al.[18] reported lower physical well-being in participants on antihypertensive medication. Boal et al.[68] reported differential effects of antihypertensive medication on risk of hospital admissions for mood disorders. Additionally, intake of antidepressant drugs has been previously related to elevated blood pressure[16]. To account for these potentially confounding factors, we investigated effects of the presence or absence of a lifetime major depression diagnosis, other forms of chronic illness as well as intake of (anti-hypertensive) medications in sensitivity analyses. Importantly, the findings of our study were robust and consistent: Independent of any potential confounders including medication intake, there was a mood-lifting effect of higher SBP on depressive symptoms and well-being, as well as a negative effect of HTN diagnosis on mood.

The results reported here are contingent on several limitations. We used UK Biobank data which is not representative of the middle-aged and older UK population[69]. The sample, particularly the neuroimaging sub-sample, displays the healthy volunteer effect, which describes that UK Biobank participants are considered to be more health-conscious, self-reported fewer health conditions and show lower rates of all-cause mortality and total cancer incidence compared to the general population[69]. Yet, associations between risk factors and disease outcomes in UK Biobank have been reported to be generalisable despite the healthy volunteer effect[69,70].

Similar to previous studies, we used self-reports of HTN and antihypertensive medications. Self-reported HTN can underestimate the true underlying prevalence[71] and may influence subjective health itself[51]. In addition, participants' self-reports of prescribed antihypertensive medications might overestimate the actual number of medications taken due to poor adherence[72]. In addition to self-report measures, we included direct standardised blood pressure recordings, as well as Likert-scale measures of depressive symptoms and well-being, which enabled us to parametrically model these exposures and outcomes in linear regression models. The mental health assessments were, however, not designed for psychiatric diagnostics, which leaves the question open whether the observed effects manifest in the sub-clinical and/or clinical range of psychiatric symptoms.

A recent study also showed that averaging blood pressure values from the first and second reading, as we did here, might underestimate the true prevalence of HTN[73]. While we have not used the blood pressure readings to define HTN in our study, we acknowledge that the procedure of blood pressure readings may have an undetected effect on our results. Yet, our results converge also when using HTN diagnosis from self-reports and hospital records, which strengthens the overall confidence in the robustness of our findings.

Conceptually, one may question the strict dichotomy between HTN versus no HTN, particularly as blood pressure thresholds for HTN diagnosis have shifted towards lower values in the last decades. However, given the diagnostic criteria for HTN at the time of the study, we assume them to be followed by most physicians in clinical care. Thus, we consider the self-reported HTN status to be a reasonable definition with predictive value for clinical outcomes in an epidemiological study[72].

Based on our preregistered hypotheses, we only tested linear associations between blood pressure and mental health. While Montano[16] showed that non-linear models testing cross-sectional blood pressure-mood associations do not outperform linear models, longitudinal trajectories of both blood pressure, mental health and their interaction over the lifespan are plausible and may reveal diverging patterns. Furthermore, observed effects' sizes were small and showed point-to-group-level, rather than individual-level patterns. Thus, no direct conclusion can be drawn from our observations relevant for individual patient care. Given the known effects of blood pressure on emotional processing, we hypothesise that – despite the small inter-individual effect sizes in our study – more pronounced intra-individual effects might exist. Our study may motivate future work testing the hypothesis that blood pressure variations and associated mental health need to be taken into account, also in the individual management of people at risk for

HTN. Future studies should also address the point of potential practical implications for treating high blood pressure in severe clinical depression. Considering the high prevalence of HTN and its treatment in the general population, as well as rising numbers of sub-clinically elevated blood pressure, small effect sizes may be epidemiologically relevant.

Our fMRI findings may be confounded by alterations in neuro-vascular coupling with HTN[74]. While this should be largely accounted for by reporting the BOLD activation *contrast* between two different types of stimuli (emotional faces versus shapes), we cannot exclude the remaining impact of impaired neurovascular coupling. Since the Hariri-task was limited to faces with negative emotions, future studies may add information about the impact of blood pressure on the neural processing of positive emotions.

Importantly, our results are not ideal to draw firm conclusions about the causality and directionality of the associations between blood pressure, HTN, and mental health, particularly the differentiation between effects of blood pressure per se and HTN remaining complex. Future longitudinal studies should therefore include earlier baseline assessments, repeated and/or continuous blood pressure monitoring over long time periods combined with frequent assessments of mental health and neuroimaging. Finally, randomised controlled trials targeted at assessing the bi-directional relationships of blood pressure and mental health will provide strong designs to elucidate these effects.

In summary, in a large British population sample of generally healthy middle-aged and older individuals, we found a relationship of elevated blood pressure with fewer depressive symptoms and greater well-being extending to a follow-up period up to around 10 years. We found the opposite effect for diagnosed HTN – already years before the timepoint of diagnosis. Participants who were normotensive at baseline and later developed HTN showed alterations in the blood pressure-mental health relationship already at baseline. An additional strength of our study lies in the use of fMRI analyses, which suggested an impact of blood pressure levels and HTN development on the neural processing of emotional stimuli. The results were overall robust to bias of medications, chronic illness, survival, social factors, and unmeasured confounds. While the observed effects are small and results from this observational study may not be directly applicable to clinical outcomes, our study adds perspective on how the interrelation of sub-clinical mental health and blood pressure might be involved in blood pressure increases during ageing and development of HTN that could have implications for developing new preventive and therapeutic approaches.

## Methods

This study was conducted using data from the UK Biobank (http://www.ukbiobank.ac.uk/) to investigate cross-sectional and longitudinal associations between blood pressure, hypertension, and mental health in behaviour and brain. The UK Biobank received ethical approval from the North West Multi-centre Research Ethics Committee (MREC; reference 11/NW/0382) in 2011, with renewed approval every 5 years since. All participants gave written informed consent. This approval means that researchers do not require separate ethical clearance and can operate under the Research Tissue Bank (RTB) approval. We were granted access to UK Biobank's data resource in May 2018 after an initial application, but embargoed data access until we completed the preregistrations of this and related studies (date of initial data release 12 Feb 2019, preregistration of this study: https://osf.io/638jg/). All analyses and data visualisations were performed with R (4.0.2, R Core Team, 2015, Vienna, Austria; R-project.org/) and RStudio (version 2022.07.0). To allow for replication studies and reproducibility of our results, the analysis code can be found in the Open Science Framework repository of this study (https://osf.io/v3yxd/).

### UK Biobank study design

The UK Biobank is a publicly available, on-going longitudinal study that aims to comprehensively assess the health-related indices of more than 500,000 UK citizens[30]. UK Biobank included participants from the UK between the ages of 40 and 69 years at recruitment in 2006 to 2010. The size of the cohort was determined based on statistical power calculations for nested case-control studies to achieve large incidences and reliable odds ratios of common health-related conditions during the first years of the 20-years follow-up period[30].

The initial assessment of the whole cohort was conducted between 2006-2010 in 22 assessment centres throughout the UK. The complete assessment protocol was repeated at two other instances, specifically at first repeat assessment visit (2012-2013), and at imaging follow-up which included ongoing brain magnetic resonance imaging (MRI) assessments of 100,000 UK Biobank participants that has started in 2014[75,76]. In addition, a sub-sample of the baseline cohort participated in an online mental health follow-up assessment in 2016. A set of questions and measures designed for the assessment of current and lifetime mental health and psychosocial factors were administered at several instances during UK Biobank data acquisition[40,77,78]. For this study, we used data from the initial assessment visit (i.e., the baseline visit), the online mental health follow-up at around 5 years from baseline, as well as the follow-up imaging visit (i.e., the follow-up visit) at around 10 years from baseline (Fig. 1). The following variables were included in this study (details on the UK Biobank data fields that were used for each variable are reported in Supplementary Table 1).

### Outcome variables

Current depressive symptoms: During each assessment centre visit, frequency of current depressive symptoms in the last two weeks (i.e., depressed mood, unenthusiasm, tenseness, tiredness) was assessed using a 4-point Likert scale ranging from 0 ("not at all") to 3 ("nearly every day"). The item scores were summarized as mean scores for the purpose of this study. Questions regarding current depressive symptoms were assessed via a touchscreen during baseline and follow-up assessment centre visits[40,77,78].

We also preregistered analyses using the Patient Health Questionnaire 9-question version (PHQ-9)[79] from the online mental health follow-up assessment which a sub-sample of the whole UK Biobank cohort received[40]. In the online PHQ-9 questionnaire, the severity of current depressive symptoms in the last two weeks was assessed. Participants indicated presence of recent depressive symptoms on a 4-point Likert scale ranging from 0 ("not at all") to 3 ("nearly every day"). Since UK Biobank applied a different coding scheme, the items were recoded to match the original coding as described above. PHQ-9 symptom scores were summarized as a sum score. The analyses and results of this questionnaire were almost identical to the ones obtained from the main depressive symptoms outcome measure described above.

Well-being: Seven questions addressing different aspects of participants' well-being were included at the assessment centre visits. The questions asked how happy/satisfied participants were regarding their happiness, and health, work, family, friendship, and financial situation, respectively. Participants were asked to respond to the questions on a 6-point Likert scale ranging from 1 ("extremely happy") to 6 ("extremely unhappy"). For the purpose of this study, all item scores were recoded and summarized as mean scores with higher scores representing greater well-being.

### Predictor variables

Systolic blood pressure (SBP): Systolic and diastolic blood pressure readings were taken from all UK Biobank participants. At each assessment centre visit, two readings were recorded with an automated Omron 705 IT electronic blood pressure monitor (OMRON Healthcare Europe B.V. Kruisweg 577 2132 NA Hoofddorp). The two

blood pressure readings were obtained during seated resting periods interleaved with a verbal interview on demographic and health-related data by UK Biobank staff. The procedure was conducted in the following sequence: (1) Interview Part 1 (demographic data), (2) First measurement of blood pressure, (3) Interview Part 2 (diseases and medications), (4) Measurement of Pulse Wave Velocity, (5) Second measurement of blood pressure. A timer ensured the second blood pressure reading could not be taken until at least 1 minute had elapsed. We used the mean of the two readings to derive systolic (SBP) values per participant at baseline and follow-up visit.

Hypertension diagnosis (HTN): Complementary to blood pressure readings, we used a categorical measure of HTN diagnosis. Participants indicated on a touchscreen whether a doctor has ever told them that they have had high blood pressure. These self-reports were given at each assessment visit and used in this study as an assessment if a diagnosis of HTN was present (yes/no) at baseline and follow-up.

Number of antihypertensive medications: In a nurse interview at each assessment visit, participants reported all medications they were currently taking. A physician from our team (LL) examined these medication lists and identified all antihypertensive drug classes (Supplementary Fig 2). We subsequently coded these antihypertensives and calculated a sum score for each participant, indicating the number of antihypertensive drugs taken. A higher number of antihypertensive medications suggests that different drug classes are required to counteract the effects of blood pressure dysregulation and thus serves as an indicator of HTN severity.

### Additional variables used in exploratory analyses

Definition of hypertension: In the moderation analysis, we aimed to explore the relevance of blood pressure-mental health associations in relation to HTN development. For this purpose, our approach was to define HTN as conservatively as possible to avoid (1) missing hypertensive cases and (2) confounding influences of antihypertensive intake. Hence, we defined hypertension for this analysis as either having a HTN diagnosis (as described above) or using any antihypertensive medication. For the moderation analysis, this definition was used for exclusion of participants with hypertension at baseline to select a non-hypertensive sample at baseline. At follow-up, this definition was used to define which participants became hypertensive between baseline and follow-up (i.e., having a HTN diagnosis at follow-up or taking antihypertensives at follow-up) and who stayed normotensive.

Imaging-derived phenotypes (IDPs): For analyses relating to brain function, we utilised the imaging-derived phenotypes (IDPs) available in UK Biobank which have been generated by processing the raw neuroimaging data. Magnetic resonance imaging (MRI) was performed at 3 Tesla and details of the imaging protocols and data processing procedures have been described in detail elsewhere[75,76]. In brief, the functional MRI task that was implemented in UK Biobank is the Hariri "emotion" task[31,32]. Participants were presented with blocks of trials and asked to decide either which of two faces presented on the bottom of the screen match the face at the top of the screen, or which of two shapes presented at the bottom of the screen match the shape at the top of the screen. The faces had either an angry or fearful expression. Trials were presented in blocks of 6 trials of the same condition (faces or shapes), with the stimulus presented for 2000ms and a 1000 ms inter-trial interval. Each block was preceded by a 2000ms task cue ("shape" or "face"), so that each block was 21 seconds including the cue. Each of the two runs included 3 face blocks and 3 shape blocks, with 8 seconds of fixation at the end of each run. IDPs of the Hariri task reflect the strength of response to the stimuli within a given brain region using the blood oxygenation level dependent (BOLD) contrast. BOLD fMRI relies on regional differences in haemoglobin oxygenation (more precisely local concentration changes of deoxy-haemoglobin) to measure regional brain activity[80–82]. Here, we focused on the IDPs of

significant clusters in amygdala and whole-brain activation for the faces > shapes contrast (median z-statistics of BOLD activation), which reflects the emotional brain response.

### Covariates

To adjust for confounding factors, we included the following variables as covariates in the analyses: age, gender, body mass index (BMI), resting heart rate, diabetes diagnosed by doctor (yes/no), lifetime depression diagnosed by doctor (yes/no), angina diagnosed by doctor (yes/no), myocardial infarction diagnosed by doctor (yes/no). Note that we refer to the variable *gender* and not sex, as the UK Biobank defines this variable as a mixture of the sex the NHS had recorded for the participant and self-reported sex, although we acknowledge that this does not capture the full spectrum of gender.

### Statistical methods

We performed cross-sectional and longitudinal multiple linear regression models. For cross-sectional comparisons, we used data of the initial assessment visit. Outcomes were current depressive symptoms and well-being, respectively. SBP, HTN and the number of antihypertensives were entered simultaneously as predictors. For longitudinal comparisons, we predicted outcome measures (i.e., depressive symptoms and well-being) assessed at the follow-up visit from predictors assessed at the initial assessment visit (i.e., SBP, HTN and the number of antihypertensives). The analyses were performed for the total sample, as well as for a subset of participants with HTN diagnosis. All statistical models included the same covariates from initial assessment visit (specified above, i.e., age, gender, BMI, resting heart rate, history of diabetes, angina, myocardial infarction, and lifetime depression). Missing data were listwise excluded and outcomes, predictors and covariates were z-scored. In all models, we have also assessed any potential influence of multicollinearity by evaluating the Variance Inflation Factor (VIF). The VIFs never exceeded a value of 2, which indicates that multicollinearity is low and inferences from our models are likely not biased due to correlations among the variables included.

### Exploratory analysis I: Moderation of the BP-mental health relationship by the development of hypertension

Using moderation analysis, we explored whether the relationship between SBP and mental health was dependent on hypertension status at follow-up assessment. For this analysis, we only included participants who were *not* hypertensive at baseline, as defined by a HTN diagnosis or intake of antihypertensive medications at this timepoint. We then compared the relationship between SBP and mental health in participants who developed hypertension (i.e., received a hypertension diagnosis or started taking antihypertensives) until the follow-up assessment, with those who stayed normotensive. The moderation analysis was performed through a multiple linear regression model including depressive symptoms or well-being as outcomes, respectively, and the interaction term of SBP at initial visit and hypertension at follow-up as predictor[83]. We also included the following covariates: age at initial visit, gender, and BMI at initial visit.

### Exploratory analysis II: Associations with emotion-related brain function

Complementary to self-reports of mental health, we explored the prospective effects of elevated blood pressure on emotion-related brain function as a central nervous outcome measure of emotional reactivity. As described in the introduction, we would expect blood pressure elevations to relate to altered neural processing of affective stimuli due to baroreceptor effects. Measures of brain function included BOLD fMRI activation in significant clusters in the amygdala and whole-brain analyses (see section above).

We used these measures as outcomes in two-sided t-tests comparing groups of participants with and without HTN, as well as in multiple linear regression models, which included SBP at follow-up (i.e., imaging visit), as well as age, gender, and BMI as covariates. In the linear regression models, we further included the interaction of SBP and HTN diagnosis at follow-up as predictor to test if the relationship between blood pressure levels and emotional reactivity differs between individuals who became hypertensive and those who did not (see also section above *Moderation of the BP-mental health relationship by development of hypertension*).

## Reporting summary

Further information on research design is available in the Nature Portfolio Reporting Summary linked to this article.

## Data availability

All data used in this study is available through the public resource of the UK Biobank (http://www.ukbiobank.ac.uk/). The raw UK Biobank data are protected and are not available due to data privacy laws. The UK Biobank data are available under restricted access to bona fide researchers for health-related research in the public interest. Access can be obtained by a registration and application to the UK Biobank resource. Data-field IDs of UKB variables used in this study are listed in the Supplementary Material. The authors' access to the UK Biobank Resource was granted under Application Number 37721 (https://www.ukbiobank.ac.uk/enable-your-research/approved-research/understanding-the-role-of-the-brain-and-cardiovascular-factors-in-hypertension-affect-relationships). Source data for figures are provided with this paper. Source data are provided with this paper.

## Code availability

To allow for replication studies and reproducibility of our results, the analysis code can be found in the Open Science Framework repository of this study (https://osf.io/v3yxd/), which links to the accompanying github repository (https://github.com/linaschaare/bp_mood).

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

## Acknowledgements

This research has been conducted using the UK Biobank Resource under Application Number 37721. We would like to sincerely thank all participants and staff who made it possible that this rich resource is available to the scientific community.

## Author contributions

L.S., M.B., D.K., M.U., and A.V. conceptualized the study. L.S., M.B., D.K., M.U., and A.V. designed and developed the methodologies. L.S. analyzed, visualized, and curated the data, as well as managed the research project. L.L. curated the medication data. L.S. and A.V. wrote the initial draft of the manuscript. A.V. and S.L.V. supervised the research. All authors critically reviewed and edited the manuscript and contributed to the final version of the paper.

## Funding

## Competing interests

The authors declare no competing interests.
