## [Peer Review File · Nature Communications]

Associations between mental health, blood pressure and the development of hypertensionEditorial Note: This manuscript has been previously reviewed at another journal that is not operating a transparent peer review scheme. This document only contains reviewer comments and rebuttal letters for versions considered at *Nature Communications*.

REVIEWER COMMENTS

Reviewer #1 (Remarks to the Author):

The authors have addressed satisfactorily most of the Reviewers previous comments. The outstanding issues are:

Firstly, the use of repeated assessment data on 20,000 of the original 500,000 as a follow-up which is not entirely appropriate (particularly given the substantial differences between participants and non-participants). This limitation is clearly substantial and its implication for the findings discussed in detail - a more rigorous estimation would be to use the 5-year mental health follow-up data as the main longitudinal analyses.

Secondly, the rationale for modeling the systolic blood pressure and HTN is unclear - the two items are clearly similar conceptually and imply one another (HTN definition is based on SBP values), raising concerns around multicollinearity.

Finally, joint modeling of SBP and survival time for hypertension patients is one alternative approach that could account for competing risk (see Kosa & Erango, 2021, Scientific Reports). Alternatively, the two measures could be included as time-varying covariates.

Reviewer #1 (Remarks to the Author):

The authors have addressed satisfactorily most of the Reviewers previous comments. The outstanding issues are:

Firstly, the use of repeated assessment data on 20,000 of the original 500,000 as a follow-up which is not entirely appropriate (particularly given the substantial differences between participants and non-participants). This limitation is clearly substantial and its implication for the findings discussed in detail - a more rigorous estimation would be to use the 5-year mental health follow-up data as the main longitudinal analyses.

We would like to thank the Reviewer for their critical assessment of the data at hand. We agree that the sample of the ongoing follow-up, that we received access to, is substantially smaller (and different) than the data in the 5-year online mental health follow-up. Interestingly, despite the differences between the two groups, the results from both follow-up assessments yielded virtually identical results (Supplementary Figure 4) which, to us, is indicative of the robustness of the findings, above and beyond selection bias. In response to the Reviewer's comment, we now highlight the 5-year mental health follow-up data in more detail in the main manuscript: In addition to reporting the complete longitudinal findings from the online mental health follow-up in detail in the Supplement and referring to them in the main manuscript (Methods, p. 24-26; Results p. 19), we furthermore mention/discuss both analyses in the Introduction and in the Discussion section and adapted Figure 1 to include the online mental health follow-up. We believe these edits give readers all information to gain a comprehensive picture of the estimations and draw attention to both complementary analyses. The suggested changes in the manuscript are outlined below the next paragraph.

Given that the results of the 5-year online mental health follow-up were so similar to the results of the 10-year follow-up visit, we suggest leaving the latter in the main text and describing the 5-year follow-up in more detail in the Supplement. There are two additional reasons for our suggestion: First, using the same set of variables in both the cross-sectional and longitudinal models provides consistency in measurements and reporting of standardised betas. This allows readers to directly compare results at baseline and follow-up. Second, this also allows us to report the results from the pre-registered well-being variable (not available at online follow-up) which we aimed to analyse in direct comparison with the depressive symptoms measure. Results from both variables are consistent and in line with pre-registered hypotheses, which we believe is important to transparently report.

Introduction (p. 3)

“The UK Biobank combines a deeply phenotyped longitudinal cohort with high statistical power of more than 500,000 participants (Sudlow et al., 2015) which enables the detection of robust small effects.

In addition, it includes two follow-up timepoints for longitudinal analyses in two sub-samples of the

baseline cohort: an online mental health follow-up at around 5 years and a follow-up at around 10 years.

We hypothesized for cross-sectional and longitudinal analyses that increased blood pressure relates to fewer depressive symptoms and greater well-being (preregistration: <https://osf.io/638jg/>).

Discussion (p. 15)

“In this study, we confirmed two seemingly contradictory associations of high blood pressure with mental health: (i) Higher SBP was associated with fewer depressive symptoms and greater well-being at the initial exam as well as at the 5-year mental health online follow-up and the 10-year-follow-up including imaging, whereas (ii) the presence of a HTN diagnosis was associated with greater depressive symptoms and lower well-being.”

Revised Figure 1:

Figure 1 – Overview of study design, outcome and predictor variables and analyses.

Secondly, the rationale for modeling the systolic blood pressure and HTN is unclear - the two items are clearly similar conceptually and imply one another (HTN definition is based on SBP values), raising concerns around multicollinearity.

We would like to thank the Reviewer for this comment and the opportunity to clarify the rationale for using both SBP and HTN as predictors in the models.

From a conceptual perspective, the “puzzling” finding of our study is that systolic blood pressure *per se*, on the one hand, and current (or future) HTN diagnosis on the other hand, have opposite relationships to well-being and depressive symptoms. While disentangling the two parameters will always be challenging, we have to include them in one model to identify such differential effects, given that some evidence for differential effects has been reported earlier (e.g., Herrmann-Lingen et al., *Psychosom Med*, 2018; Montano, *J Psychophysiol*, 2019; Licht et al. *Hypertension*, 2009). As the reviewer indicated, methodological considerations of collinearity are very important. Interestingly, from a methodological perspective, we did not observe high interdependence (multicollinearity) to bias statistical inferences (i.e. standard errors / confidence intervals) in our regression models. Indeed, the correlation of SBP and HTN at baseline in our data is $r = 0.34$. This effect is small to medium and according to simulation studies would not introduce issues of multicollinearity in our regression models (O'Brien, 2017, <https://doi.org/10.1111/ssqu.12273>). In all models, we have also assessed any potential influence of multicollinearity by evaluating the Variance Inflation Factor (VIF), which reflects the ratio of the variance of a regression coefficient in a model that includes multiple other parameters by the variance of a model constructed using only one parameter. For all variables in the models, the VIF never exceeded a value of 2 (e.g. in cross-sectional models: $VIF_{SBP} = 1.28$, $VIF_{HTN} = 1.73$), which supports that multicollinearity was low and inferences from our models are likely not biased due to correlations among the variables included (Mansfield & Helms, 1982, [10.1080/00031305.1982.10482818](https://doi.org/10.1080/00031305.1982.10482818); See also discussion in Allison, 2012, <https://statisticalhorizons.com/multicollinearity/>). We included information on the VIFs in the Methods section under *Statistical methods* (p. 28):

“In all models, we have also assessed any potential influence of multicollinearity by evaluating the Variance Inflation Factor (VIF). The VIFs never exceeded a value of 2, which indicates that multicollinearity is low and inferences from our models are likely not biased due to correlations among the variables included.”

In response to the Reviewer’s comment, we furthermore include a statement regarding the complexity of disentangling the effect of blood pressure and HTN diagnosis in the discussion section of the manuscript. We also suggest that future studies should investigate the relationship between blood pressure, HTN, and emotional well-being probably by combining long term continuous BP monitoring with frequent assessments of well-being. We added this point in the Discussion (p. 21):

“Importantly, our results are not ideal to draw firm conclusions about causality and directionality of the associations between blood pressure, HTN and mental health, with particularly the differentiation

between effects of blood pressure *per se* and HTN remaining complex. Future longitudinal studies should therefore include earlier baseline assessments, repeated and/or continuous blood pressure monitoring over long time periods combined with frequent assessments of mental health and neuroimaging. Finally, randomised controlled trials targeted at assessing the bi-directional relationships of blood pressure and mental health will provide strong designs to elucidate these effects.”

Finally, joint modeling of SBP and survival time for hypertension patients is one alternative approach that could account for competing risk (see Kosa & Erango, 2021, Scientific Reports). Alternatively, the two measures could be included as time-varying covariates.

We appreciate the Reviewer elaborating on the implementation of such models for this study. As the Reviewer suggested, we modelled the effects of SBP and HTN in a joint model using Linear Mixed Effects Models (LMM). LMMs enable a joint approach to model time-varying effects of the variables, while accounting for their covariance structures across repeated measures and in the presence of unbalanced designs (e.g. due to drop-outs). The results of this analysis coherently confirm the multiple regression results reported in the main manuscript. Notably, we observed a significant effect of SBP on mental health over time, in line with the previous results indicating a relationship between SBP and depressive symptoms that changes between assessments with a steeper negative slope at baseline than at follow-up. The analysis is reported in the Supplementary Materials and referred to in the main manuscript (p. 10):

Results (p. 10)

“Additional and sensitivity analyses

We performed several additional analyses to test the robustness of these results. Additional analyses included i) using the PHQ-9 questionnaire of the 5-year online mental health follow-up as a validated instrument to assess current depressive symptoms, ii) additional relevant covariates, such as socioeconomic status, insomnia, racial/ethnic background, insomnia, etc., iii) using hospital inpatient data for diagnoses of HTN and depression, iv) assessment of potential survival bias, v) exploration of potential unmeasured confounding effects with E-values and vi) joint modelling of time-varying effects of SBP and HTN using Linear Mixed Effects Models. Moreover, sensitivity analyses were performed to test whether the above reported results were dependent on vii) the presence or absence of previous diagnosis of depression or any other severe disease that might affect BP (list of diseases in

Supplementary Materials); viii) the intake of antidepressants or any other medication intake; ix) a specific effect of certain antidepressant or antihypertensive drug classes.”

Supplementary Materials (p. 20-21):

“Joint time-varying analysis using Linear Mixed Effects Models

To directly model time-varying effects of SBP and HTN in a joint model that accounts for the unbalanced design in the follow-up measures, we explored the longitudinal association with mental health in a Linear Mixed Effects Model framework using R’s *lmer* function. Separate models were set up for depressive symptoms and well-being as outcomes, respectively. The full models further included a random intercept per participant, the same covariates of no interest as in the main analyses, fixed effects for SBP, HTN and number of antihypertensive medications, as well as three two-way interactions: SBP*HTN, SBP*time point and HTN*time point. Time point was coded as a dummy variable for “initial assessment” and “imaging follow-up”. Each full model was compared to a null model that was identical but did not include the interactions. The difference between the full and the null model was tested using the *anova* function and setting the argument *test* to “chisq” to perform a Chi-squared test. Chi-squared values and p-values for each interaction effect were derived by dropping interactions iteratively (reduced models). Non-significant interactions were dropped from the full model to yield a less complex reduced model.

For depressive symptoms, the full-null-model comparison was significant ($X^2(3) = 10.67, p = 0.014$). Iterative model reductions revealed that the interaction of SBP*time point was significant ($X^2(1) = 10.55, p = 0.001$), while HTN*timepoint ($X^2(1) = 1.78, p = 0.182$) and HTN*SBP ($X^2(1) = 0.11, p = 0.974$) were not significant. The significant interaction of SBP and time point indicates that the relationship between SBP and depressive symptoms changes between assessments. Evaluation of the fixed effects revealed that the interaction was driven by a steeper negative slope at baseline than at follow-up. The stronger negative relationship between SBP and depressive symptoms at baseline compared to follow-up supports our observation from multiple linear regression analyses showing that the relationship was stronger for the cross-sectional analysis at baseline than for the longitudinal analysis

including follow-up outcomes. As expected, HTN diagnosis showed no significant interaction with time point, but a significant positive main effect ($X^2(1) = 358.69, p < 0.001$) indicating that the effect of HTN diagnosis on depressive symptoms was stable across assessments. No significant difference between individuals with and without HTN was found regarding the relationship between SBP and depressive symptoms across both time points (i.e., no significant SBP*HTN interaction). This observation is in line with the results from the moderation analysis, which showed a moderating effect of impending HTN on the relationship between SBP and depressive symptoms at baseline assessment, but not at follow-up. For well-being, the full-null-model comparison was not significant ($X^2(3) = 5.39, p = 0.145$). While we did not find significant interaction effects on well-being - suggesting that the main effects remained constant over time - the positive effect of SBP and negative effect of HTN on well-being reported in the main manuscript was corroborated by significant main effects of these variables (HTN: $X^2(1) = 256.54, p < 0.001$, SBP: $X^2(1) = 352.86, p < 0.001$). With these additional mixed model analyses, we were able to include all available data including participants with missing data; confirming the findings from multiple linear regression models reported in the main manuscript.”

REVIEWER COMMENTS

Reviewer #1 (Remarks to the Author):

I am satisfied with the additional analyses performed by the authors and the responses to my earlier concerns. This Reviewer has three remaining concerns:

1. Would it be feasible to run a quintile regression for the association between SBP and depression?
2. Would it be feasible to repeat the baseline cross-sectional association with a similar one using one of the follow-up data? This would verify the robustness of the association, minimising the risk of spurious association.
3. Would be of interest to discuss whether, based on the presented evidence, practitioners should avoid controlling and managing high BP in patients with both depression and hypertension in order to enhance mood symptoms, even if this means increasing the risk of major CVD events or mortality.

Thank you.

REVIEWER COMMENTS

Reviewer #1 (Remarks to the Author):

I am satisfied with the additional analyses performed by the authors and the responses to my earlier concerns. This Reviewer has three remaining concerns:

1. Would it be feasible to run a quintile regression for the association between SBP and depression?

We would like to thank the Reviewer for this interesting suggestion and have added the following evaluation of the binned association between SBP and depression to the Supplementary Materials (p. 21-22):

“Association of depressive symptoms within systolic blood pressure categories

We additionally evaluated how the association between SBP and depression presents in varying quantiles along the range of SBP levels, similar to previous studies (Rapsomaniki et al., 2014; Su et al., 2014; Yano et al., 2018).

We divided the data into 7 bins following the commonly used categories for diagnostic SBP classification (“<90” [Reference Category], “90-120”, “120-130”, “130-140”, “140-160”, “160-180”, “>180” mmHg). Next, we computed the cross-sectional association between SBP bins and depressive symptoms at baseline while adjusting for the same covariates as in the main manuscript.

Interestingly, the results indicate fewer depressive symptoms for each unit increment in bins of SBP (Supplementary Figure 8), suggesting that SBP positively affects mood symptoms at greater rates in higher ranges of SBP levels until it decreases again in ranges of hypertensive crises (> 180 mmHg). The estimate for SBP between 90-120 mmHg was not significantly different from the reference category (estimate = -0.070, 95% CI [-0.167 0.027], p = 0.156), which supports our findings that the negative association between SBP and mood might be particularly salient for individuals at risk of high blood pressure.”

Supplementary Figure 8 - Cross-sectional associations of depressive symptoms within systolic blood pressure categories at initial assessment. Forest plot shows regression estimates and 95% confidence intervals for each systolic blood pressure category and the respective sample size in each bin. SBP <90 mmHg (n = 109) served as the reference category in the model. The model has been fully adjusted for diagnosed hypertension, number of antihypertensives, and other covariates (age, gender, body mass index, resting heart rate, diabetes diagnosed by doctor (yes/no), lifetime depression diagnosed by doctor (yes/no), angina diagnosed by doctor (yes/no), myocardial infarction diagnosed by doctor (yes/no)). Total sample n = 303,771 (after exclusion of missing values).

2. Would it be feasible to repeat the baseline cross-sectional association with a similar one using one of the follow-up data? This would verify the robustness of the association, minimising the risk of spurious association.

We would like to thank the Reviewer for this suggestion addressing their concern about robustness of the results. Following the suggestion, we repeated the baseline cross-sectional association at both follow-up timepoints and included the results, which confirm the main results, in the Supplementary Materials (p. 22-23):

“Cross-sectional association of mental health outcomes at both follow-up assessments

We further analysed cross-sectional models at both follow-up time points. Despite smaller sample sizes, resulting in larger 95% confidence intervals, the results yielded robust, significant estimates for SBP in line with our previous analyses (Supplementary Figure 9). Estimates for HTN diagnosis showed that the direction of effects was positive for depressive symptoms ($\beta = 0.007$; 95% CI [-0.017, 0.030]; $p = 0.579$) and negative for well-being ($\beta = -0.016$; 95% CI [-0.038, 0.007]; $p = 0.183$) at 10-year follow-up, consistent with the analyses in the main manuscript, yet, results were not significant. For the number of antihypertensive medications, we observed significant associations with all mental health outcomes, suggesting that higher hypertension burden, indicated by a higher number of prescribed antihypertensives, negatively relates to mood.”

Supplementary Figure 9 - Cross-sectional associations of mental health outcomes at both follow-up assessments. Forest plot shows standardized beta estimates and 95% confidence intervals for predictors of interest (systolic blood pressure, diagnosed hypertension (HTN), and number of antihypertensives) as well as covariates at 10-year follow-up. N = 10,333 for current depressive symptoms at 10-year follow-up, n = 8,122 for depressive symptoms at 6-year online mental health follow-up, and n = 10,990 for well-being at 10-year follow-up (after exclusion of missing values).

3. Would be of interest to discuss whether, based on the presented evidence, practitioners should avoid controlling and managing high BP in patients with both depression and hypertension in order to enhance mood symptoms, even if this means increasing the risk of major CVD events or mortality.

We thank the Reviewer for this suggestion and the Reviewer is right that such discussions might arise. Though the current study shows robust relationships between blood pressure variations and mood, it would be a bit premature to draw practical conclusions from our study for individual patient care, Yet, we agree with the Reviewer that it offers several interesting questions for follow-up studies to address, such as whether and how physicians treating elevated blood pressure should consider mental-health implications and specifically how to deal with blood pressure lowering in patients with major depression. In the revised manuscript, we therefore added this latter point as an additional suggestion for future studies (p. 21):

“Furthermore, observed effects’ sizes were small and showed point to group-level, rather than individual-level patterns. Thus, no direct conclusion can be drawn from our observations relevant for individual patient care. Given the known effects of blood pressure on emotional processing, we hypothesise that – despite the small inter-individual effect sizes in our study – more pronounced intra-individual effects might exist. Our study may motivate future work testing the hypothesis that blood pressure variations and associated mental health need to be taken into account, also in the individual management of people at risk for HTN. Future studies should also address the point of potential practical implications for treating high blood pressure in severe clinical depression. Considering the high prevalence of HTN and its treatment in the general population, as well as rising numbers of sub-clinically elevated blood pressure, small effect sizes may be epidemiologically relevant.”